# LYRA: GENERATIVE 3D SCENE RECONSTRUCTION VIA VIDEO DIFFUSION MODEL SELF-DISTILLATION

**Sherwin Bahmani**[1,2,3]   **Tianchang Shen**[1,2,3]   **Jiawei Ren**[1]   **Jiahui Huang**[1]   **Yifeng Jiang**[1]
**Haithem Turki**[1]   **Andrea Tagliasacchi**[2,4]   **David B. Lindell**[2,3]   **Zan Gojcic**[1]
**Sanja Fidler**[1,2,3]   **Huan Ling**[1]   **Jun Gao**[1*]   **Xuanchi Ren**[1,2,3*]
[1]NVIDIA  [2]University of Toronto  [3]Vector Institute  [4]Simon Fraser University
*equal contribution

https://research.nvidia.com/labs/toronto-ai/lyra

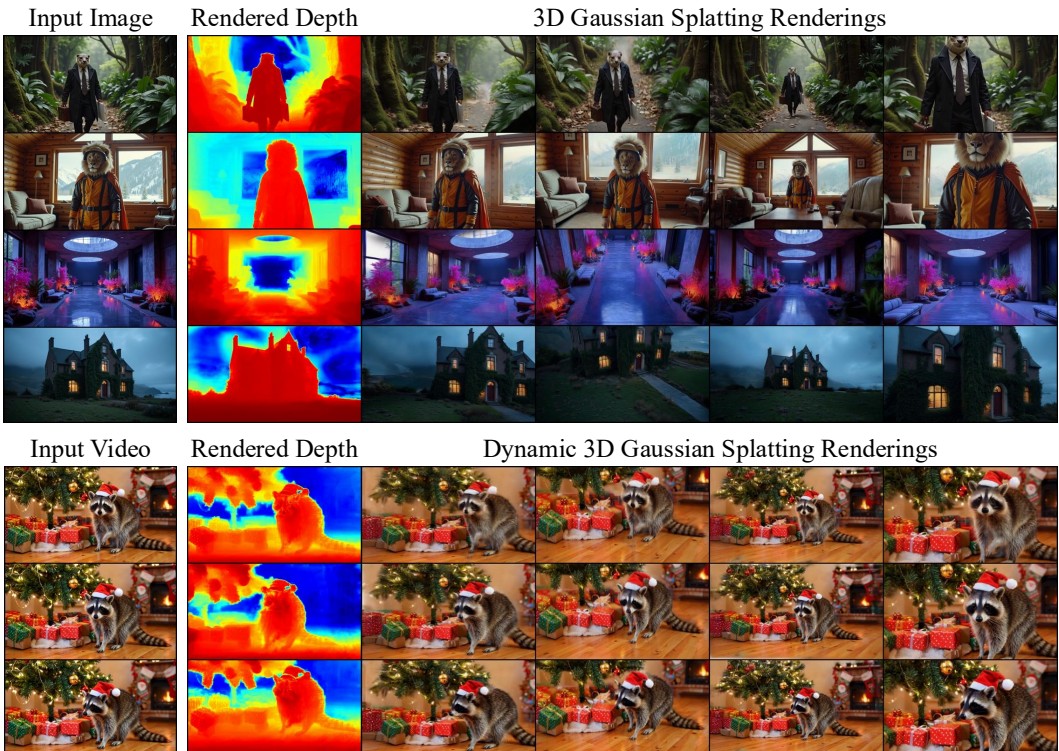

Figure 1: **Feed-Forward 3D and 4D Scene Generation.** From a single image (top), *Lyra* infers a 3D Gaussian Splatting (3DGS) representation in a feed-forward fashion, through self-distilling a video diffusion model without requiring real-world multi-view data. With a video input (bottom), *Lyra* infers a dynamic 3DGS that offers interactive control in both time (rows) and viewpoint (columns).

## ABSTRACT

The ability to generate virtual environments is crucial for applications ranging from gaming to physical AI domains such as robotics, autonomous driving, and industrial AI. Current learning-based 3D reconstruction methods rely on the availability of captured real-world multi-view data, which is not always readily available. Recent advancements in video diffusion models have shown remarkable imagination capabilities, yet their 2D nature prevents their use in simulations where a robot needs to navigate and interact with the environment. In this paper, we propose a self-distillation framework that aims to distill the implicit 3D knowledge in the video diffusion models into an explicit 3D Gaussian Splatting (3DGS) representation, eliminating the need for multi-view training data. Specifically, we augment the typical RGB decoder with a 3DGS decoder, which is supervised by the output of the RGB decoder. In this approach, the 3DGS decoder can be purely trained with synthetic data generated by video diffusion models. At inference time, our model

can synthesize 3D scenes from either a text prompt or a single image for real-time rendering. Our framework further extends to dynamic 3D scene generation from a monocular input video. Experimental results show that our framework achieves state-of-the-art performance in static and dynamic 3D scene generation. Video results: `https://research.nvidia.com/labs/toronto-ai/lyra`

# 1 INTRODUCTION

Creating high-quality 3D environments at scale is a long-standing challenge in computer vision and graphics, empowering applications across film-making, VR/AR, and physical AI with closed-loop simulation. These applications require explicit 3D representations that support real-time rendering, physical interaction, and consistent multi-view synthesis.

Recent advances in neural 3D reconstruction (Mildenhall et al., 2020; Kerbl et al., 2023; Moenne-Loccoz et al., 2024) enable recovering such representations from posed images. However, the reliance on accurate camera poses and high-quality images significantly limits their scalability. The challenge is even greater for dynamic scenes, which typically require synchronized multi-camera setups (Li et al., 2022). Reconstruction methods are also inherently constrained to the observed content and cannot extrapolate beyond the input views. Recent works aim to mitigate these limitations by using feed-forward reconstruction models (Zhang et al., 2024e; Ren et al., 2024c), but these approaches face their own bottleneck: the scarcity of diverse, large-scale 3D training data, which leads to poor out-of-domain generalization.

Video diffusion models (Cosmos, 2025; Wan, 2025) offer a promising alternative. Trained on massive internet-scale video corpora, they achieve impressive fidelity and generalization across diverse environments. By learning from real-world videos with varied camera trajectories, ranging from handheld motion to cinematic panning and drone footage, these models implicitly encode cues about the underlying 3D world without requiring multi-view training data. Unlike reconstruction approaches (Charatan et al., 2024), generative models can also hallucinate plausible content beyond what is visible in the input frames. Yet, video diffusion models generate only 2D frames, lacking explicit 3D representations. This limits their use in simulation and downstream tasks that demand geometric consistency, long-term coherence, and physical interaction. Encouragingly, recent work (Wang et al., 2024e; Ren et al., 2025) has shown that video models can be adapted for explicit camera control using relatively small-scale posed datasets. This ability to generate posed image sequences transforms them into a powerful tool that can serve as both input and supervision for 3D reconstruction models.

Motivated by this insight, we bridge these two paradigms, reconstruction and generation, through what we call *generative 3D scene reconstruction*. We introduce *Lyra*, a novel method for generating explicit 3D environments from the latent representations of video diffusion models in a single forward pass. Cruically, *Lyra* is trained in a self-distillation framework (Fig. 2), with a 3D Gaussian Splatting (3DGS) decoder operating directly in the latent space of a video diffusion model. Given a single input image or video, we sample a camera trajectory as conditioning input, denoise the resulting video latent, and decode it along two parallel branches. The first branch uses the standard RGB decoder to synthesize a video sequence, while the second is our 3DGS decoder, which produces an explicit 3D representation. Together, these branches form a self-distillation framework in which the RGB branch (teacher) supervises the 3DGS branch (student). To extend viewpoint coverage, we sample multiple camera trajectories, generate multiple video latents, and train the 3DGS decoder to fuse information across them while mitigating long-term and multi-view inconsistencies.

This generative reconstruction framework provides three key benefits: **(i)** it enables generation of large-scale synthetic environments spanning diverse scenarios directly from video diffusion models, removing the need for real-world multi-view captures; **(ii)** by operating in latent space of a video model, it allows efficient processing of multiple views without the heavy memory overhead of pixel-space feed-forward reconstruction methods; and **(iii)** its explicit 3DGS output guarantees geometric consistency, providing representations that are directly applicable to downstream tasks such as physical simulation. Consequently, at inference time, *Lyra* generates high-quality 3D Gaussian scenes from monocular input that support real-time rendering, *without requiring any additional optimization or post-processing*.

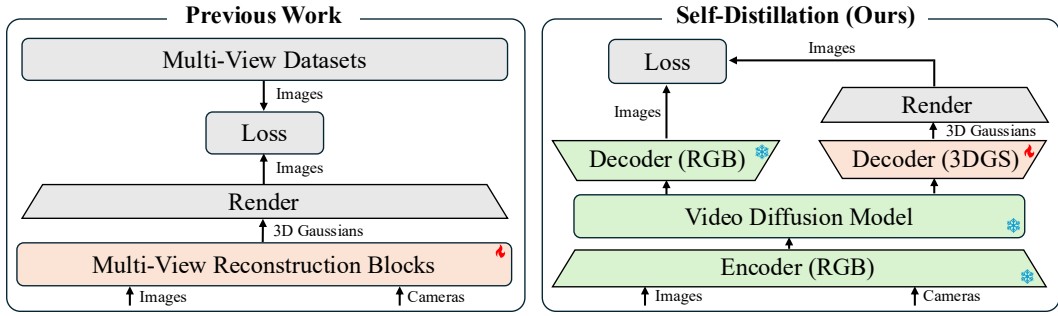

Figure 2: **Self-distillation framework of *Lyra*.** Previous work (left) (Szymanowicz et al., 2025b) trains multi-view reconstruction blocks using real-world datasets with limited diversity (Zhou et al., 2018; Ling et al., 2024b). In contrast, we propose a self-distillation framework (right) for generative 3D scene reconstruction. Precisely, a pre-trained camera-controlled video diffusion model with its RGB decoder output (teacher) supervises the rendering of the 3DGS decoder (student). Using a video model pre-trained on diverse 2D video data allows us to provide diverse multi-view supervision.

We further extend this self-distillation framework to dynamic 4D generation from monocular video. In this setting, the video model (teacher) provides space–time supervision, while the student learns to produce time-conditioned 3DGS representations that enable novel-view synthesis of dynamic scenes.

Overall, our work makes the following contributions:

- We introduce a self-distillation framework that trains a 3DGS student decoder using a pre-trained camera-controlled video diffusion model as the teacher, eliminating the need for captured multi-view real-world data.

- Our framework extends to dynamic scenes to support 4D reconstruction from a monocular video.

- Our model generalizes across diverse scenes, achieving state-of-the-art results in single-image 3D scene generation and single-video 4D scene generation.

## 2 RELATED WORK

**Multi-view image generation.** Early works on multi-view image generation (Watson et al., 2023; Liu et al., 2023; Shi et al., 2024; Wang & Shi, 2023) mainly focus on object-centric scenes without background. Follow-up works (Sargent et al., 2024; Tang et al., 2023b; Schneider et al., 2025) extend multi-view image generators to the scene scale. CAT3D (Gao et al., 2024b) extends a pre-trained image diffusion model for multi-view image generation from any generated or real image input. In a subsequent stage, the generated multi-view images are reconstructed into an explicit 3D representation, e.g., NeRF (Mildenhall et al., 2020) or 3DGS (Kerbl et al., 2023). Moreover, these methods have been extended to generate dynamic 3D scenes from multiple viewpoints using a space-time diffusion model (Watson et al., 2024; Wu et al., 2025b; Kuang et al., 2024; Wang et al., 2025a). Even though these methods demonstrate impressive results, grounding the generations in an explicit 3D representation requires an expensive optimization stage that is not amortized across scenes. Instead of creating multi-view images and then reconstructing using optimization, we are interested in *directly* generating a 3D scene from text or a single image.

**Camera-conditioned video generation.** There has been significant progress in fine-tuning video diffusion models for 3D camera control. MotionCtrl (Wang et al., 2024e) pioneered camera control by conditioning pre-trained video models with camera poses. Follow-up works (He et al., 2024a; Xu et al., 2024c; Bahmani et al., 2025a;b) represent cameras as Plücker coordinates for pixel-wise conditioning. Another line of work (Hu et al., 2024; Hou et al., 2024; Zhang et al., 2024b) investigates training-free camera control of video diffusion models. Recently, SynCamMaster (Bai et al., 2025b) and ReCamMaster (Bai et al., 2025a) demonstrated impressive camera-controlled video synthesis using large-scale synthetic training data. While these works achieve compelling results, their output is not grounded in 3D. Our work is orthogonal to this line of work, as we tackle the task of feed-forward 3D generation from camera-controlled video models, allowing novel viewpoint rendering *after* generation. Concretely, we build upon GEN3C (Ren et al., 2025), a recent camera-controlled video diffusion model, and use it as a teacher within a self-distillation framework for 3D reconstruction.

**Feed-forward 3D models.** Early work on feed-forward 3D reconstruction mainly focused on object-centric scenes without background. These methods directly predict a NeRF (Hong et al., 2024; Li et al., 2024b) or 3DGS (Tang et al., 2024a; Szymanowicz et al., 2024) from either text or an image. More recent works (Charatan et al., 2024; Zhang et al., 2024e; Szymanowicz et al., 2025a; Ren et al., 2024c) focus on scene-level 3D reconstruction, typically regressing per-pixel 3D Gaussians from one or more input images. However, most scene-level 3D reconstruction methods are limited to training distribution scenes, e.g., RealEstate10K (Zhou et al., 2018), with limited generalizability to generated scenes. Recent methods extend feed-forward 3D reconstruction models to generate scenes. Bolt3D (Szymanowicz et al., 2025b) trains a pointmap (Wang et al., 2024b) autoencoder to generate pointmaps along the multi-view images, used for feed-forward 3D reconstruction. Closely related to our work, Wonderland (Liang et al., 2025a) uses a camera-controlled video model to generate 3D Gaussians with a feed-forward network. In contrast to these works, we use a camera-controlled video model as a teacher within a self-distillation framework to train our student 3D model *without requiring real-world multi-view data*. Importantly, we show a simple extension of our framework for feed-forward 4D scene generation, an unexplored task to date.

## 3 SELF-DISTILLATION USING VIDEO DIFFUSION MODELS

Our key idea is to distill the implicit 3D knowledge embedded in video diffusion models into an explicit 3DGS decoder capable of producing high-quality 3D representations. To this end, we build a teacher–student framework in which the video diffusion model (teacher) generates RGB videos that supervise the 3DGS decoder (student) that operates in the same latent space, as illustrated in Fig. 2. In this section, we detail our video diffusion model backbone (Sec. 3.1) and its role in self-distillation for the 3DGS decoder (Sec. 3.2). We discuss the design choice for the 3DGS decoder in Sec. 4 and outline minimal changes required to adapt the pipeline to dynamic 3D scenes in Sec. 5.

### 3.1 BACKGROUND: CAMERA-CONTROLLED AND 3D-CONSISTENT VIDEO DIFFUSION

Since we supervise our 3DGS decoder only with the camera-controlled video diffusion model, we heavily rely on its 3D consistency. Due to this, we build our approach on GEN3C (Ren et al., 2025), a recent camera-conditioned video diffusion model. In the following we provide an overview of the key design choices made in GEN3C.

**Spatiotemporal 3D cache.** To improve video consistency and camera control precision, GEN3C constructs a spatiotemporal 3D cache $\{\mathbf{P}^{t,v}\}$ from input image(s) or videos, where each $\mathbf{P}^{t,v}$ is a colored point cloud obtained by unprojecting the depth estimation (Wang et al., 2024a) of an RGB image captured from camera viewpoint $v$ at time $t$. The cache is organized as an $L \times V$ array, where $L$ denotes the number of frames (temporal length) and $V$ denotes the number of camera views.

**Rendering and structured guidance.** To leverage this cache, GEN3C renders each point cloud from arbitrary given camera poses, producing RGB images $\mathbf{I}^{t,v}$ and disocclusion masks $\mathbf{M}^{t,v}$ via $(\mathbf{I}^{t,v}, \mathbf{M}^{t,v}) = \mathcal{R}(\mathbf{P}^{t,v}, \mathbf{C}^t)$, where $\mathcal{R}$ denotes the rendering function that projects the 3D point cloud $\mathbf{P}^{t,v}$ onto the 2D camera plane according to the camera pose $\mathbf{C}^t$ at time $t$. Given a sequence of camera poses $\mathbf{C} = (\mathbf{C}^1, \ldots, \mathbf{C}^L)$, rendering all cache elements produces $V$ videos of length $L$, which can be stacked into image sequences $\mathbf{I}^v \in \mathbb{R}^{L \times 3 \times H \times W}$ and mask sequences $\mathbf{M}^v \in \mathbb{R}^{L \times 1 \times H \times W}$ for each view $v$. The disocclusion masks indicate areas that the video diffusion model should fill in. These renderings serve as structured visual guidance for subsequent video generation.

**Video variational autoencoder (VAE).** In diffusion-based video generation models, a video variational autoencoder (VAE) (Kingma & Welling, 2013; Rombach et al., 2022) is commonly employed to compress videos into a lower-dimensional latent space for efficient training and inference. Given a RGB video $\mathbf{I} \in \mathbb{R}^{L \times 3 \times H \times W}$, a pre-trained VAE encoder $\mathcal{E}$ will encode the video into a latent space, i.e. $\mathbf{z} = \mathcal{E}(\mathbf{I}) \in \mathbb{R}^{L' \times C \times h \times w}$. The training and inference of the diffusion model are then performed in this latent space. The final video $\hat{\mathbf{I}} = \mathcal{D}_{rgb}(\mathbf{z})$ is decoded with a pre-trained VAE decoder $\mathcal{D}_{rgb}$. In this paper, we adopt the pre-trained GEN3C (Ren et al., 2025; Cosmos, 2025) model. Specifically, the latent channel dimension is $C = 16$, the temporal dimension is $L' = (L-1)/\tau + 1$, and the spatial dimensions are $h = H/\sigma$ and $w = W/\sigma$, where the temporal compression factor is $\tau = 8$ and the spatial compression factor is $\sigma = 8$.

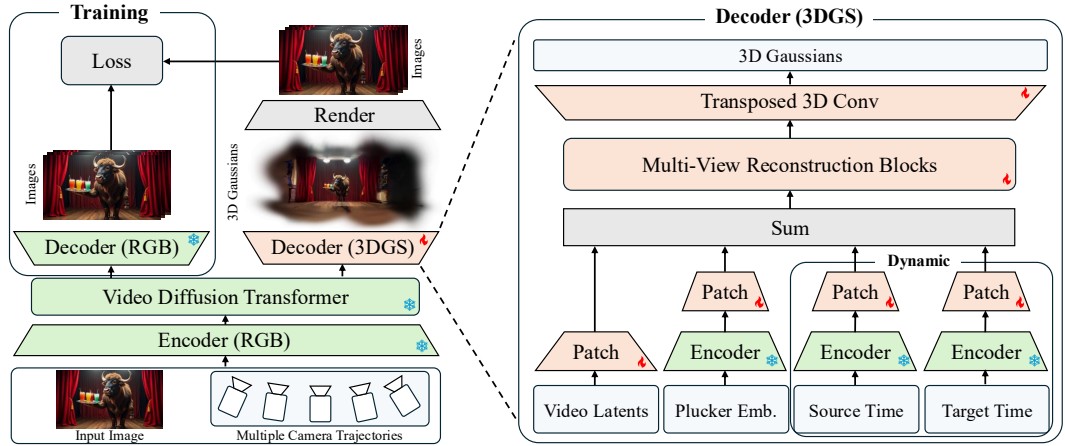

Figure 4: **3D Generative Reconstruction Framework.** Our pipeline builds upon a camera-controlled video diffusion model (Ren et al., 2025) pre-trained on large scale data. We train a 3D Gaussian Splatting (3DGS) decoder by aligning the 2D image renderings of generated 3DGS scenes with the RGB-decoded generations of the pre-trained video model. We only train the 3DGS decoder while freezing the pre-trained autoencoder and diffusion model. At inference time we directly use the 3DGS decoder, without requiring the RGB decoder anymore. Time conditioning within the 3DGS decoder allows us to easily extend our approach from static to dynamic 3D scene generation.

## 3.2 SELF-DISTILLATION

We train the 3DGS decoder $\mathcal{D}_s$ as a student under a teacher–student paradigm, where a camera-controlled video diffusion model $\mathcal{V}$ serves as the teacher. For diverse supervision, we curate a large-scale set of diverse text prompts with large language models (OpenAI, 2025; Bai et al., 2023), generate images $I$ with an image diffusion model (Black Forest Labs, 2024), and expand into multi-view sequences using GEN3C (Ren et al., 2025).

**Teacher–student setup.** Given an input image $I$ and a sampled camera trajectory $\{\mathbf{C}^t\}_{t=1}^L$, the video diffusion model $\mathcal{V}$ generates a denoised video latent $\mathbf{z} = \mathcal{V}(I, \{\mathbf{C}^t\}_{t=1}^L)$. The latent $\mathbf{z}$ is decoded along two branches: the pre-trained RGB decoder $\mathcal{D}_{rgb}$ produces video frames $\mathbf{I}_{\mathcal{D}_{rgb}} = \mathcal{D}_{rgb}(\mathbf{z})$, while the 3DGS decoder $\mathcal{D}_s$ outputs explicit 3D Gaussians $\mathbf{G}$. Rendered views from 3DGS *(student)* are defined as $\mathbf{I}_{\mathcal{D}_s} = \mathrm{Render}(\mathbf{G}, \{\mathbf{C}^t\}_{t=1}^L)$ and are supervised to match RGB frames $\mathbf{I}_{\mathcal{D}_{rgb}}$ *(teacher)*, forming the self-distillation loop.

**Multi-trajectory supervision.** In our paper, to enlarge viewpoint coverage for the input image, we sample $V = 6$ camera trajectories per input image, as shown in Fig. 3. For each trajectory $v$, we construct a spatiotemporal cache $\{\mathbf{P}^{t,v}\}_{t=1}^L$ along with $L = 121$ camera poses $\{\mathbf{C}^{t,v}\}_{t=1}^L$. Passing these caches through video diffusion model $\mathcal{V}$ yields latents $\mathbf{z}^v$, which the teacher decodes into RGB frames $\mathbf{I}_{\mathcal{D}_{rgb}}^v = \mathcal{D}_{rgb}(\mathbf{z}^v)$. The 3DGS decoder $\mathcal{D}_s$ learns to fuse these multiple $\mathbf{z}^v$ into coherent Gaussians $\mathbf{G}$, while filling disoccluded regions. Through this self-distillation design, the 3DGS decoder $\mathcal{D}_s$ is trained entirely from synthetic supervision provided by $\mathcal{V}$. Operating in latent space enables efficient aggregation of multiple trajectories, while the explicit Gaussian output $\mathbf{G}$ enforces geometric consistency and supports downstream simulation and real-time rendering.

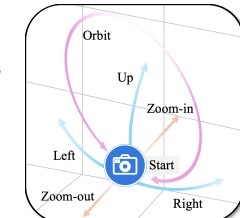

Figure 3: Sampled six camera trajectories to maximize view coverage.

## 4 FEED-FORWARD RECONSTRUCTION FROM MULTI-VIEW VIDEO LATENTS

Our feed-forward 3DGS decoder $\mathcal{D}_s$ is designed to transform the synthesized multi-view latents generated by our video model $\mathcal{V}$ into an explicit 3D representation that can be rendered from arbitrary viewpoints. We choose 3D Gaussian Splatting (3DGS) (Kerbl et al., 2023) due to its explicit representation and fast rendering speed. We visualize our detailed 3DGS decoder in Fig. 4 (right).

## 4.1 3DGS Decoder

**Scaling Latent-Based 3D Reconstruction.** Previous feed-forward reconstruction frameworks generate 3D Gaussians for each pixel, and thus, are limited by the number and resolution of input views they can handle. For example, GS-LRM (Zhang et al., 2024e) operates on 2–4 images at $512 \times 512$ resolution, while AnySplat (Jiang et al., 2025a) is trained with 24 images at $448 \times 448$ resolution. Our video model synthesizes $V \times L = 6 \times 121 = 726$ input views at $704 \times 1280$ resolution—far beyond the capacity of existing methods, which exceeds GPU memory limit. The bottleneck lies in the attention mechanism applied to visual tokens, whose memory and compute requirements grow with the number of pixels. To overcome this limitation, we avoid scaling in pixel space and instead operate directly in the compressed video latent space produced by our camera-controlled video diffusion model. The multi-view video latents are denoted as $\mathbf{Z} \in \mathbb{R}^{V \times L' \times C \times h \times w}$.

**Architecture.** The 3DGS decoder $\mathcal{D}_s$ first maps its inputs into the hidden dimension of the main reconstruction blocks and outputs per-pixel 3D Gaussian features $\mathbf{G} \in \mathbb{R}^{V \times L \times H \times W \times 14}$. Specifically, the inputs are the multi-view video latents $\mathbf{Z}$ and encoded Plücker embeddings $\mathbf{E}$, both with latent dimensions $\mathbf{Z}, \mathbf{E} \in \mathbb{R}^{V \times L' \times C \times h \times w}$. Each input component is first patchified to match the hidden dimension of the main reconstruction blocks. We do not require any additional visual encoder—only a spatial $2 \times 2$ patchification layer (Dosovitskiy et al., 2021) to transform the video latents into flattened tokens for the reconstruction network. The sum of both inputs is processed through the reconstruction blocks. We follow the design introduced in Long-LRM (Ziwen et al., 2024), i.e., one block consists of one Transformer (Vaswani et al., 2017) layer followed by seven Mamba-2 (Dao & Gu, 2024) layers. We repeat the block twice to get 16 layers with 512 hidden dimensions. Finally, a transposed 3D convolution maps the hidden representation to 14 Gaussian channels: 3D position $(x, y, z)$, scale $(s_x, s_y, s_z)$, rotation quaternion $(q_w, q_x, q_y, q_z)$, opacity $\alpha$, and RGB $(r, g, b)$. Formally, the static decoder is expressed as $\mathbf{G} = \mathcal{D}_s(\mathbf{Z}, \mathbf{E})$.

**Plücker embeddings.** Raw Plücker embeddings are first computed from camera poses $\{\mathbf{C}^{t,v}\}$ and intrinsics as $\mathbf{E}^{\text{raw}} \in \mathbb{R}^{V \times L \times 6 \times H \times W}$. We reuse the RGB encoder $\mathcal{E}$ from the pre-trained VAE to encode Plücker embeddings. Specifically, we separately encode the 3-dimensional ray directions and the 3-dimensional cross product of ray directions and origins using $\mathcal{E}$, and concatenate latent along the channel dimension to obtain $\mathbf{E}^{\text{enc}} \in \mathbb{R}^{V \times L' \times 2C \times h \times w}$. A lightweight MLP maps $\mathbf{E}^{\text{enc}}$ to $\mathbf{E} \in \mathbb{R}^{V \times L' \times C \times h \times w}$, reducing the channel dimension by a factor of 2 to match video latent $\mathbf{Z}$.

## 4.2 Loss function

**Image-based supervision.** We supervise our 3DGS decoder with an image-based reconstruction loss. The reconstruction loss is split into a Mean Squared Error (MSE) loss $\mathcal{L}_{mse}$ and an LPIPS loss $\mathcal{L}_{lpips}$ (Zhang et al., 2018) using VGG (Simonyan & Zisserman, 2014) as a feature extractor.

**Depth supervision.** We observed that the decoder trained with only RGB loss often produces flattened geometry. To address this, we additionally supervise the rendered depth maps using the consistent video depth estimated by an off-the-shelf system ViPE (Huang et al., 2025), and the scale-invariant depth loss $\mathcal{L}_{depth}$ from Long-LRM (Ziwen et al., 2024).

**Opacity-based pruning.** Similar to Long-LRM (Ziwen et al., 2024), we use an L1 regularization on the opacity $\mathcal{L}_{opacity}$ and remove the Gaussians with the lowest 80% opacity.

**Total loss.** Our total loss is computed as the weighted sum of all losses with weight factors $\lambda_i$

$$\mathcal{L} = \lambda_{mse}\, \mathcal{L}_{mse} + \lambda_{lpips}\, \mathcal{L}_{lpips} + \lambda_{depth}\, \mathcal{L}_{depth} + \lambda_{opacity}\, \mathcal{L}_{opacity} \tag{1}$$

where we set $\lambda_{mse} = 1.0$, $\lambda_{lpips} = 0.5$, $\lambda_{depth} = 0.05$, and $\lambda_{opacity} = 0.1$.

## 5 Extension to Dynamic 3D Scenes

Our approach can be extended to handle dynamic 3D scenes with minimal changes, outlined below.

**Self-distillation for dynamic scenes.** Our dynamic 3D setup closely follows the design of the static 3D counterpart. Instead of a single image, the video model takes a single video of length $L$ with corresponding camera poses as input and generates multi-view video latents capturing the same underlying motion. For video inputs, we follow a protocol similar to the static case: sampling

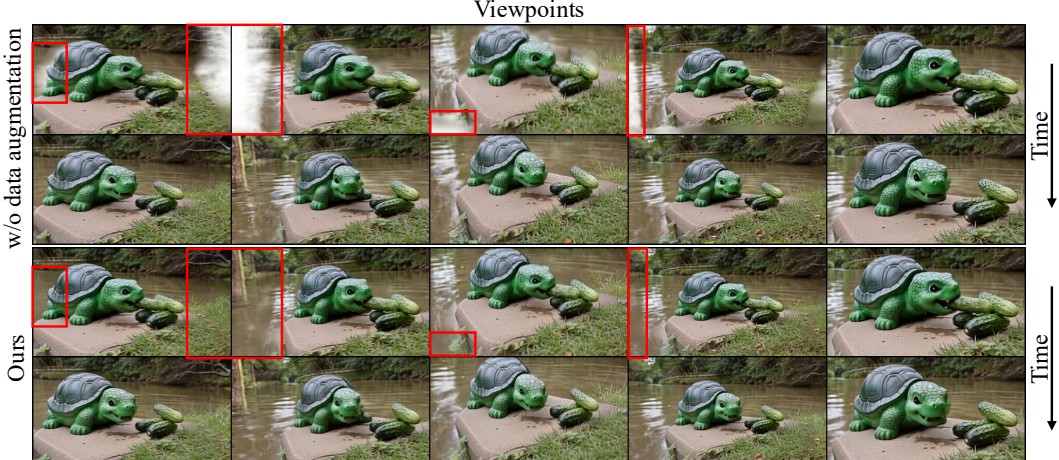

Figure 5: **Dynamic data augmentation.** When naively supervising the time-aware 3DGS decoder, we observe artifacts in the generated 3D Gaussians. Specifically, early timesteps from extreme poses exhibit low opacity in regions not covered by the supervision signal. To address this, we augment the supervision data with a motion-reversed video, ensuring that each timestep is observed from the full spatial coverage, thereby preventing low opacity artifacts in the early timesteps.

diverse text prompts with large language models, then generating videos using video diffusion models (Cosmos, 2025; Wan, 2025). The generated videos are then annotated with camera poses and depth maps using ViPE (Huang et al., 2025).

**Dynamic 3DGS decoder architecture.** We follow the bullet-time design of Liang et al. (2025b) and design our dynamic 3DGS decoder $\mathcal{D}_s$ to generate time-dependent 3D Gaussians for all the video frames. Specifically, the dynamic 3DGS decoder $\mathcal{D}_s$ closely follows the architecture of $\mathcal{D}_s$, except that we augment the $\mathcal{D}_s$ input with the encoded source and target time embeddings $\mathbf{T}^{\text{src}}, \mathbf{T}^{\text{tgt}} \in \mathbb{R}^{V \times L' \times C \times h \times w}$. The unencoded (raw) source times are assigned to each input frame, forming $\mathbf{T}^{\text{src,raw}} \in \mathbb{R}^{V \times L \times 1 \times H \times W}$, and the target time is $\mathbf{T}^{\text{tgt,raw}} \in \mathbb{R}^{V \times 1 \times 1 \times H \times W}$. Both are spatially repeated and augmented with a 2-dimensional sinusoidal embedding to expand the channel dimension to 3, then encoded with the RGB encoder $\mathcal{E}$ to latent dimensions $\mathbf{T}^{\text{src}}, \mathbf{T}^{\text{tgt}} \in \mathbb{R}^{V \times L' \times C \times h \times w}$. The target time is repeated along the latent temporal dimension to match $L'$. During training, a target time step is randomly sampled, and the corresponding 3D Gaussians are decoded and supervised across all trajectories generated with dynamic data augmentation. We fine-tune $\mathcal{D}_d$ from a pre-trained $\mathcal{D}_s$ with the patchification layers of $\mathbf{T}^{\text{src}}$ and $\mathbf{T}^{\text{tgt}}$ initialized to zeros. $\mathbf{T}^{\text{src}}$ and $\mathbf{T}^{\text{tgt}}$ are directly added to the sum of $\mathbf{Z}$ and $\mathbf{E}$. Formally, the dynamic decoder is defined as $\mathbf{G} = \mathcal{D}_d(\mathbf{Z}, \mathbf{E}, \mathbf{T}^{\text{src}}, \mathbf{T}^{\text{tgt}})$.

**Dynamic data augmentation.** In the static 3D scenes, where time is not a factor, frames from different timesteps can all be used to supervise the 3DGS renderings. In contrast, for dynamic 3D scenes, the 3DGS changes over time, so only frames from the corresponding timestamp are valid for supervision. This restriction can lead to a trivial solution, where the 3DGS at a given timestep simply ignores information from frames at other timesteps (see Fig. 5).

To mitigate this issue, we introduce a dynamic data augmentation strategy that balances supervision across timesteps. Early frames are naturally associated with viewpoints close to the input image, whereas later frames correspond to more distant viewpoints. To counter this imbalance, we augment the training data with paired supervision views for each timestep — one near and one far. Concretely, we reverse the input video in frame order and feed it into the video model $\mathcal{V}$, which produces six additional multi-view sequences. After reversing these sequences back to the original motion order, we obtain six trajectories that progress inward (from far viewpoints toward the input). Combined with the original six outward trajectories, this yields 12 supervision views per timestep. Importantly, this augmentation is applied only during training: the reversed trajectories act purely as extra supervision signals to prevent collapse from pruning artifacts and are not required at inference time.

Table 1: **State-of-the-art comparisons.** We compare our method with previous works for single image to 3D generation using RealEstate10K, DL3DV, and Tanks-and-Temples datasets.

| Method | RealEstate10K | | | DL3DV | | | Tanks-and-Temples | | |
|---|---|---|---|---|---|---|---|---|---|
| | PSNR ↑ | SSIM ↑ | LPIPS ↓ | PSNR ↑ | SSIM ↑ | LPIPS ↓ | PSNR ↑ | SSIM ↑ | LPIPS ↓ |
| ZeroNVS | 13.01 | 0.378 | 0.448 | 13.35 | 0.339 | 0.465 | 12.94 | 0.325 | 0.470 |
| ViewCrafter | 16.84 | 0.514 | 0.341 | 15.53 | 0.525 | 0.352 | 14.93 | 0.483 | 0.384 |
| Wonderland | 17.15 | 0.550 | 0.292 | 16.64 | 0.574 | 0.325 | 15.90 | 0.510 | 0.344 |
| Bolt3D | 21.54 | 0.747 | 0.234 | - | - | - | - | - | - |
| Ours | **21.79** | **0.752** | **0.219** | **20.09** | **0.583** | **0.313** | **19.24** | **0.570** | **0.336** |

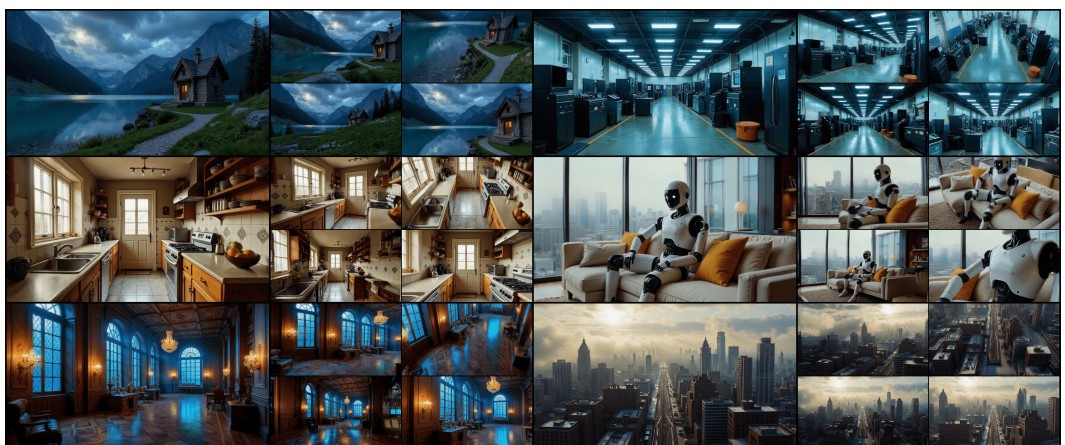

Figure 6: **Image-to-3DGS Generation.** We visualize five views from generated 3DGS scenes.

# 6 EXPERIMENTS

## 6.1 EXPERIMENTAL SETUP

**Datasets.** To train our model, we do not use any existing multi-view datasets; instead, we construct our own, which we call the *Lyra* dataset, relying solely on our video model to supervise the 3DGS decoder. The 3D reconstruction setup uses 59,031 images, while the 4D setup has 7,378 videos. All data are from diverse text prompts, spanning scenarios such as indoor and outdoor environments, humans, animals, and both realistic and imaginative content. We synthesize six camera trajectories for each image (3D) or video (4D), yielding 354,186 videos for 3D and 44,268 videos for 4D.

**Baselines.** We compare our method with the state-of-the-art approaches for generative single-image-to-3D reconstruction, i.e., ZeroNVS (Sargent et al., 2024), ViewCrafter (Yu et al., 2024b), Wonderland (Liang et al., 2025a), and Bolt3D (Szymanowicz et al., 2025b). Note that *no source code is available* to evaluate these methods on our out-of-distribution evaluation set, hence we mainly rely on reported quantitative comparisons from the papers.

**Evaluation.** We follow previous works to evaluate our model on the task of single-image to 3D using RealEstate10K (Zhou et al., 2018), DL3DV (Ling et al., 2024b), and Tanks and Temples (Knapitsch et al., 2017). We follow the evaluation protocol outlined in Wonderland (Liang et al., 2025a) and Bolt3D (Szymanowicz et al., 2025b). We evaluate the performance using standard reconstruction metrics, i.e., PSNR, SSIM, and LPIPS.

## 6.2 MAIN RESULTS

**Quantitative results.** We show quantitative evaluations for single image-to-3D in Tab. 1. Our method outperforms previous works on all benchmarks across all metrics. Hence, the main improvements to further boost the quality lie in the development of stronger video generative models, as our reconstructions will directly benefit from them. We provide additional quantitative comparisons on the *Lyra* dataset in Appendix D.

| Ours | w/o self-distillation | w/o multi-view fusion | w/o LPIPS loss |
|---|---|---|---|

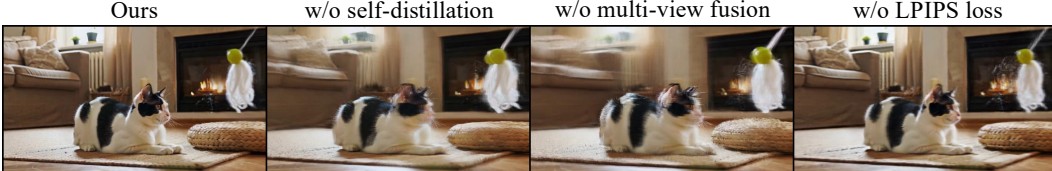

Figure 7: **Ablations.** We visualize ablation results by rendering the same extreme novel viewpoint after image-to-3DGS generation; depth visualizations and ablations are provided in Appendix D.4.

**Qualitative results.** We visualize novel view renderings from our generated 3DGS scenes in Fig. 1 and Fig. 6, but strongly recommend the reader to our supplementary webpage for video results. Our method generates high-quality novel view content for 3D/4D scenes in unseen regions while maintaining consistency with the input image/video. We visualize qualitative comparisons on the *Lyra* dataset in Appendix D.

## 6.3 ABLATIONS

In Tab. 2, we motivate our design choices by ablating key components on out-of-distribution diverse prompts. We compare 3DGS renderings with videos generated by our camera-controlled video diffusion model.

**Real data only.** Instead of using self-distillation, we train a model using only real multi-view datasets, i.e., RealEstate10K (Zhou et al., 2018) and DL3DV (Ling et al., 2024b). These datasets are commonly used in previous works, but lead to limited generalizability to out-of-distribution scenes.

**Self-distillation + real data.** Perhaps surprisingly, joint training with self-distillation and real data does not improve over using only self-distillation without real data. This confirms that self-distillation is diverse and consistent enough to learn a reconstruction model.

**No depth loss.** Depth loss prevents flat geometry and even slightly improves image-based metrics. We visualize depth maps in Appendix D.4.

**No opacity pruning.** We keep all Gaussians instead of pruning them by opacity. Learning to prune most of the 3D Gaussians makes the output representation more compact and slightly improves visual quality. Rendering at $H = 704$, and $W = 1280$ takes 18ms with pruning vs. 30ms without pruning, a $1.67\times$ speed up.

Table 2: **Ablation study on *Lyra* dataset.**

| Method | PSNR ↑ | SSIM ↑ | LPIPS ↓ |
|---|---|---|---|
| Ours | 24.77 | 0.837 | 0.224 |
| **Data** | | | |
| real data only | 19.08 | 0.659 | 0.413 |
| self-distill. + real data | 24.74 | 0.823 | 0.236 |
| **Loss** | | | |
| w/o depth loss | 24.31 | 0.811 | 0.247 |
| w/o opacity pruning | 24.55 | 0.820 | 0.237 |
| w/o LPIPS loss | 23.74 | 0.766 | 0.370 |
| **Architecture** | | | |
| w/o multi-view fusion | 17.73 | 0.632 | 0.446 |
| w/o Mamba-2 | 24.58 | 0.818 | 0.241 |
| w/o latent 3DGS | | OOM | |

**No LPIPS loss.** Adding the LPIPS loss improves the robustness to input inconsistencies and enhances the preservation of high-frequency details.

**No multi-view fusion.** A naive implementation would be to generate 3D Gaussians for each camera trajectory independently and then fuse the 3D Gaussians into one point cloud. However, we observe significant advantages by learning the multi-view fusion from different trajectories with our reconstruction blocks, where each token attends to the others.

**No Mamba-2.** We replace our joint Transformer/Mamba-2 with Transformer-only blocks and observe slightly lower quality. One forward pass with $V = 6$, $L = 121$, $H = 704$, and $W = 1280$ takes 3213ms with joint blocks in comparison to 20922ms with Transformer-only blocks, a $6.5 \times$ speedup.

**No latent-based 3DGS.** Previous works operating in the pixel space, such as BTimer (Liang et al., 2025b), are restricted to 12 input frames, while we can take up to 726 input frames. Consequently, operating the 3DGS decoder in the pixel space instead of the latent space leads to out-of-memory.

## 7    CONCLUSION

In this work, we propose *Lyra*, a novel 3D and 4D generation framework relying only on a single image or video input. Instead of collecting multi-view datasets, we introduce camera-controlled video diffusion models as teachers within a self-distillation framework for a student 3DGS decoder. Our 3DGS decoder operates in the latent space of the video model and directly reconstructs 3D Gaussians without any post-processing or optimization. This design enhances generalizability and coverage across diverse scenes, while our dynamic extension demonstrates the feasibility of 4D generation from monocular video. Currently, the scale and consistency of our generated scenes are bounded by the capacity of our camera-controlled video diffusion model. Hence, further improving the video diffusion model will enable large-scale, consistent scene synthesis. Moreover, investigating the adaptation of auto-regressive techniques (Chen et al., 2024a) into our framework for large-scale generation is an interesting direction for future work. Lastly, modelling motion and tracking information within the reconstruction network, as in concurrent work (Lin et al., 2025), could improve visual motion quality.

## 8    ETHICS STATEMENT

Like all generative AI technologies, there are risks of misuse of our method, including producing misleading or synthetic 3D/4D content. While such risks exist, the main goal of this research is to contribute to the fields of simulation, robotics, and embodied AI by providing scalable and controllable tools for data generation. In this sense, we expect our most significant contributions to be made through the following capabilities and insights.

- **The ability to enable realistic simulation for embodied agents**: We provide controllable and physically consistent camera motion within generated 3D and 4D environments to support training and evaluating agents that must perceive, navigate, and interact with complex scenes.
- **Improved scalability for data engines**: Our method removes the need for multi-view capture and expensive per-scene optimization, and helps produce high-fidelity, customizable scenes at scale, which is particularly suitable for research in robotics, reinforcement learning, and closed-loop simulation.
- **Better technical understanding**: As an academic study, this work contributes a foundational case study of how camera control interacts with spatiotemporal generative models, which will benefit research in computer vision, graphics, and embodied AI.

While we caution against deceptive or wrongful use of this technology—where misleading and/or unverifiable media could be created with realistic generative outputs—we believe that the adoption of safeguards (e.g., provenance tracking, dataset documentation, and good evaluation practices) will be useful in ensuring that generative models advance science and society in positive ways.

## 9    REPRODUCIBILITY STATEMENT

To fully reproduce our results and accelerate future research in this area, we release our training and inference code, model weights, and data for both 3D and 4D generation.

## 10    ACKNOWLEDGEMENTS

We thank Zian Wang for feedback on the draft.

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

## A    SUPPLEMENTARY WEBPAGE

We provide video results on our supplementary webpage: `https://research.nvidia.com/labs/toronto-ai/lyra`.

## B    ADDITIONAL DETAILS OF VIDEO DIFFUSION MODEL

This section provides additional information on the GEN3C (Ren et al., 2025) video diffusion model backbone introduced in Sec. 3.1 and how we further improve its 3D consistency.

**Conservative mask refinement.**  A key limitation of GEN3C's forward warping is background leakage, where occluded object parts in the source view (e.g., lion body in Fig. 8) become visible in novel viewpoints but are not properly indicated in the disocclusion mask. This leads to incomplete foreground completion by the video diffusion model, as shown in Fig. 8 (second and fourth columns).

To address this, we adopt a conservative strategy: since the geometry of occluded regions in $\mathbf{P}^{t,v}$ is unknown, we assume all such areas are occupied and derive refined disocclusion masks accordingly. This allows the video model to reason about potentially visible foreground regions that were missed by standard point-based rendering.

Our approach constructs a triangular mesh from camera-space points $\mathbf{P}_c^{t,v} = \mathbf{C}^t \mathbf{P}^{t,v}$ by connecting spatially adjacent pixels, similar to Hu et al. (2021):

$$\mathcal{M} = \bigcup_{(u,v)} \left\{ \triangle\left(\mathbf{p}_{u,v}, \mathbf{p}_{u+1,v}, \mathbf{p}_{u,v+1}\right), \triangle\left(\mathbf{p}_{u+1,v}, \mathbf{p}_{u+1,v+1}, \mathbf{p}_{u,v+1}\right) \right\} \quad (2)$$

where $\mathbf{p}_{u,v}$ denotes the 3D point at pixel $(u, v)$ in $\mathbf{P}_c^{t,v}$. This surface mesh serves as a boundary between visible and invisible regions. For each rendered pixel, we compare the standard point-based depth $\mathbf{D}^{t,v}$ with the mesh-interpolated depth $\mathbf{D}_{\mathcal{M}}^{t,v}$ obtained via ray-surface intersection:

$$\mathbf{M}^{t,v}(u, v) = \begin{cases} 0 & \text{if } \mathbf{D}_{\mathcal{M}}^{t,v}(u, v) < \mathbf{D}^{t,v}(u, v) - \epsilon \\ \mathbf{M}_{\text{orig}}^{t,v}(u, v) & \text{otherwise} \end{cases}, \quad (3)$$

where $\mathbf{M}_{\text{orig}}^{t,v}$ denotes the original disocclusion mask from forward warping and $\epsilon$ is a small tolerance factor. When the mesh surface is closer than the original points, we conservatively mask out these regions as potential foreground disocclusions. This produces more reliable guidance for video generation, especially under large viewpoint changes, as shown in Fig. 8 (third and fifth columns).

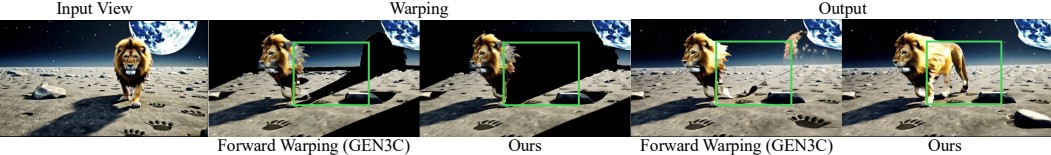

Figure 8: **Rendering function comparison.** Our model builds upon GEN3C (Ren et al., 2025) that uses forward warped images as camera control conditioning. We visualize forward warping (left) vs. our conservative rendering function (right). The highlighted region is incorrectly filled with background pixels in forward warping but properly masked in ours, enabling correct completion.

## C    MODEL DETAILS

**Progressive training.**  Directly training the 3DGS decoder on 6 trajectories of 121 frames at $704 \times 1280$ is expensive, hence we train our network progressively, as shown in Tab. 3. Furthermore, we use the final static pre-trained model to initialize our dynamic model. The total training time takes 6 days with 8 NVIDIA A100 80 GB GPUs. We use gsplat (Ye et al., 2025) as the 3DGS implementation.

**Inference.**  One forward pass through the video diffusion model takes 12 minutes. As video models become faster, especially with progress in distillation into few-step models, this number is expected

to decrease drastically with orthogonal improvements by the video generation community. Our 3DGS decoder takes only 3 seconds to generate 3D Gaussians from video latents. Lastly, we can render the 3D Gaussians in real-time after generation from novel viewpoints, where rendering a 704×1280 frame takes 18 ms.

**Efficiency and number of input trajectories.** Our approach is trained to handle a variable number of input camera trajectories. During training, we randomly vary the number of trajectories between 1 and 6, enabling the model to generalize to any number of trajectories within this range at inference time. While we primarily use 6 trajectories in our experiments to maximize view coverage, the model can operate with as few as a single trajectory.

Importantly, using fewer trajectories allows avoiding any computational overhead beyond standard camera-controlled video generation. The 3DGS decoder requires only 1 second for a single trajectory and 3 seconds for six trajectories to produce 3D Gaussians, which is comparable to decoding RGB frames in 2D video generation models. Using additional trajectories therefore provides a controllable trade-off between view coverage and inference speed, without affecting reconstruction quality when fewer trajectories are used.

**Dataset generation cost.** The training dataset is generated fully automatically, without manual screening or automated filtering, making the approach highly scalable. Dataset generation is dominated by the cost of the underlying video diffusion model, which currently takes 12 minutes per trajectory on a single NVIDIA A100 GPU. As a result, generating one training sample with 6 trajectories takes 72 minutes.

For the 3D dataset, we generate 59,031 scenes with 6 camera trajectories each, corresponding to approximately 71,000 GPU hours. For the 4D dataset, we generate 7,378 scenes with 12 trajectories each, corresponding to approximately 18,000 GPU hours. This data generation process is performed once, offline, after which any number of models can be trained on the resulting dataset.

Collecting real-world data with comparable diversity is practically infeasible, especially for the 4D case, which would require multiple cameras synchronized in time. We fully open-source the generated datasets to facilitate future research. As video generation models continue to become faster, the cost of dataset generation is expected to decrease substantially. In contrast, training the 3DGS decoder itself is relatively inexpensive, requiring 6 days on 8 NVIDIA A100 GPUs, whereas related approaches often require multi-node training.

**Subsampling Gaussians.** Generating pixel-aligned 3D Gaussians scales linearly with each spatial and temporal resolution. Our generated scenes represent 726 frames at $704 \times 1280$ resolution, leading to 654,213,120 per-pixel Gaussians. Hence, we only generate one 3D Gaussian per $8 \times 8$ spatial neighborhood, reducing the total number of 3D Gaussians by a factor of 64, i.e., 10,222,080. Moreover, our opacity-based pruning further reduces the number to 2,044,416, making the output more compact.

**Joint Transformer-Mamba blocks.** We use the reconstruction block design introduced in Long-LRM (Ziwen et al., 2024), i.e., one block consists of one Transformer layer followed by seven Mamba-2 layers. We repeat the block twice to get 16 layers with 512 hidden dimensions. Note that our network is significantly smaller than the one used in Long-LRM (Ziwen et al., 2024) or Wonderland (Liang et al., 2025a), which use 24 layers with 1024 hidden dimensions. While pure Transformers are typically more powerful, they are significantly slower in training and inference than Mamba2. Since we trained both configurations within the same training budget, we hypothesize that transformer-only blocks would need considerably more training iterations to fully catch up in rendering quality.

**Camera encoding.** We use Plücker embeddings for camera encoding, following prior works (Kant et al., 2024; Zhang et al., 2024e). The embeddings are computed at full spatial and temporal resolution and then projected into the video latent space using the pre-trained RGB encoder. Since the RGB encoder expects 3 input channels but Plücker embeddings are 6-dimensional, we separately encode the 3-dimensional ray directions and the 3-dimensional cross product of ray directions and origins. The resulting latent encodings are then concatenated along the channel dimension.

**Time encoding.** Similarly, to make the 1-dimensional time input compatible with the 3-channel input expected by the RGB encoder, we concatenate a 2-dimensional sinusoidal embedding to the original time value along the channel dimension. We then replicate these values across the spatial dimensions

Table 3: **Progressive training setup.**

| Stage | $H \times W$ | $L$ | $V$ | $S$ | $B$ | Steps |
|---|---|---|---|---|---|---|
| Static | | | | | | |
| 1 | 176×320 | 17 | 1 | 17 | 4 | 10k |
| 2 | 176×320 | 49 | 1 | 49 | 4 | 2.5k |
| 3 | 352×640 | 49 | 1 | 49 | 2 | 2.5k |
| 4 | 704×1280 | 49 | 1 | 49 | 1 | 2.5k |
| 5 | 704×1280 | 121 | 1 | 9 | 1 | 57.5k |
| 6 | 704×1280 | 121 | 1–6 | 9 | 1 | 7k |
| Dynamic | | | | | | |
| 7 | 704×1280 | 121 | 6 | 12 | 1 | 10k |

before encoding the time into the video latent space. This approach allows us to independently encode both the source and target times into the compressed latent representation.

**Dynamic data augmentation.** We visualize our dynamic data augmentation procedure for two example camera trajectories in Fig. 9. The procedure creates pairs of supervision views that cover the scene from the same motion state but different extreme viewpoints.

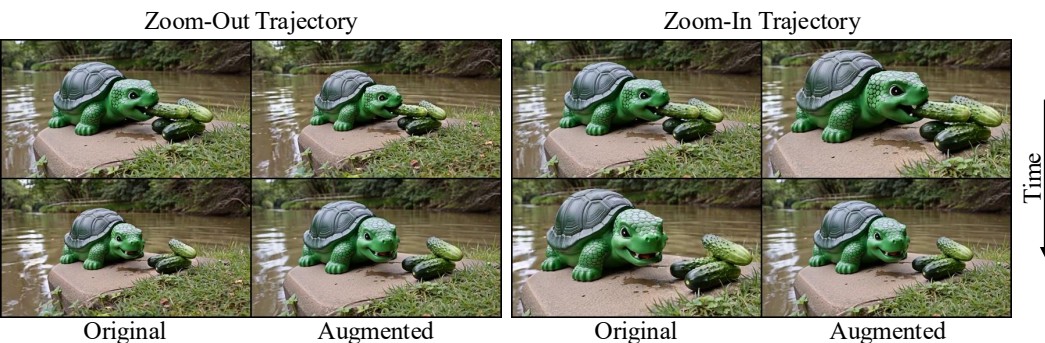

Figure 9: **Dynamic data augmentation videos.** We augment the supervision data with a motion-reversed video, ensuring that each timestep is observed from the full spatial coverage, thereby preventing low opacity artifacts in the early timesteps. We show two example trajectories, i.e., zoom-out and zoom-in, and visualize their corresponding augmented videos. The augmented videos are flipped in their camera motion.

**Multi-view consistency.** All camera trajectories are generated independently, but are rendered from a shared underlying 3D/4D cache. In our framework, the video generation backbone (GEN3C) models scene memory through this cache, enabling camera-controlled video synthesis with implicit scene consistency across trajectories.

While this shared cache already provides a degree of multi-view coherence, further improvements are possible by integrating more advanced memory mechanisms developed for long-form video generation. Recent approaches such as Context as Memory (Yu et al., 2025) explicitly retrieve and reuse scene information over extended temporal horizons, and could further strengthen multi-view consistency in a multi-trajectory setting. We leave the integration of such memory mechanisms for future work.

## D    ADDITIONAL EVALUATION

### D.1    ADDITIONAL BASELINE DETAILS

We additionally compare our model against BTimer (Liang et al., 2025b), a recent approach for joint 3D and 4D reconstruction from posed images or videos. We reached out to the authors of BTimer to conduct experiments for us using the same scenes as *Lyra*. Since BTimer is purely regression-based, we integrate it with our GEN3C (Ren et al., 2025) video diffusion backbone. Specifically, we use the

BTimer (GEN3C)               Ours

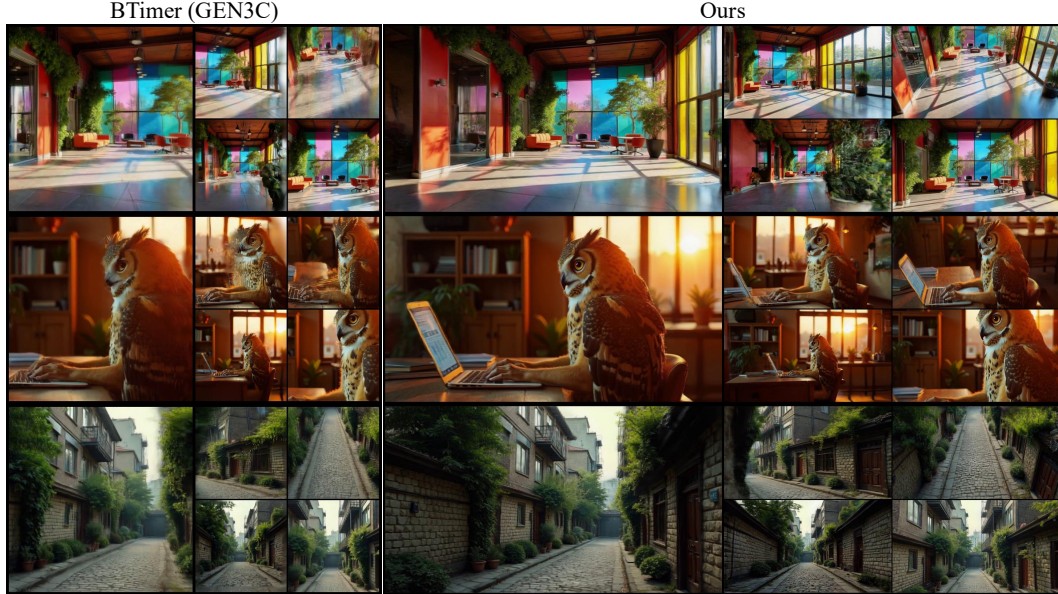

Figure 10: **Image-to-3DGS Comparison.** We compare our method with BTimer (GEN3C) and visualize five views from generated scenes. We observe significantly fewer artifacts and higher fidelity.

Table 4: **Comparison with pixel-space Long-LRM (GEN3C) baseline.** Quantitative results on RE10K, DL3DV, and Tanks&Temples datasets.

| Dataset | Method | PSNR ↑ | SSIM ↑ | LPIPS ↓ |
|---------|--------|--------|--------|---------|
| RE10K | Long-LRM (GEN3C) | 18.45 | 0.653 | 0.378 |
|       | Ours | **21.79** | **0.752** | **0.219** |
| DL3DV | Long-LRM (GEN3C) | 17.83 | 0.499 | 0.439 |
|       | Ours | **20.09** | **0.583** | **0.313** |
| T&T | Long-LRM (GEN3C) | 16.56 | 0.491 | 0.482 |
|     | Ours | **19.24** | **0.570** | **0.336** |

same camera trajectories as in our model and provide RGB-decoded videos from GEN3C as input to BTimer. BTimer operates in pixel space and requires 12 input images. However, as discussed in the main paper, pixel-space models such as BTimer run out of memory when directly using the 726 high-resolution frames generated by GEN3C. To address this, we uniformly subsample GEN3C frames. For static evaluations, we select 12 frames that evenly cover the available viewpoints. For dynamic evaluations, we sample 11 frames to span the viewpoint range and additionally include the bullet-time frame corresponding to the motion being reconstructed. This ensures that the target motion state is always present as a reference. We denote this integrated baseline as BTimer (GEN3C), i.e., BTimer combined with GEN3C.

### D.2 ADDITIONAL STATIC 3D EVALUATION

**Long-LRM + GEN3C** We compare against a pixel-space baseline that combines GEN3C with Long-LRM (Ziwen et al., 2024). In this setting, GEN3C serves as the video generation backbone, while Long-LRM performs 3D reconstruction directly in pixel space. Since Long-LRM cannot process hundreds of generated frames, we uniformly subsample the GEN3C output before using it as input. Quantitative comparisons are reported in Table 4.

**BTimer + GEN3C** For the task of image to 3DGS generation, we show qualitative comparisons with BTimer (GEN3C) in Fig. 10 and qualitative results in Tab. 6a. We observe higher quality and less artifacts for our method.

Table 5: **CLIP and IQA metrics.** Semantic alignment evaluated with CLIP (ViT-B/32, ViT-B/16, ViT-L/14; higher is better) and perceptual image quality evaluated with NIQE (lower is better).

| Method | CLIP-B/32 ↑ | CLIP-B/16 ↑ | CLIP-L/14 ↑ | NIQE ↓ |
|---|---|---|---|---|
| Long-LRM (GEN3C) | 21.10 | 20.31 | 16.43 | 19.70 |
| Ours | **32.58** | **32.38** | **27.24** | **11.34** |

Table 6: **Quantitative results on static and dynamic *Lyra* datasets.**

| Method | PSNR ↑ | SSIM ↑ | LPIPS ↓ |
|---|---|---|---|
| BTimer (GEN3C) | 16.32 | 0.580 | 0.427 |
| Ours | **24.92** | **0.834** | **0.183** |

(a) **Comparison on static *Lyra* dataset.**

| Method | PSNR ↑ | SSIM ↑ | LPIPS ↓ |
|---|---|---|---|
| BTimer (GEN3C) | 20.29 | 0.687 | 0.315 |
| Ours | **23.07** | **0.779** | **0.231** |

(b) **Comparison on dynamic *Lyra* dataset.**

**CLIP and IQA metrics.** We evaluate our method using additional semantic and perceptual quality metrics. For semantic alignment, we report CLIP scores using three variants: ViT-B/32, ViT-B/16, and ViT-L/14, where higher values indicate better text–image alignment. For image quality assessment, we compute the Naturalness Image Quality Evaluator (NIQE), where lower values correspond to better perceptual quality. Tab. 5 reports the results and compares our method with the GEN3C + Long-LRM baseline. Our approach consistently achieves higher CLIP scores across all variants and substantially lower NIQE values, indicating improved semantic consistency and perceptual image quality.

### D.3 DYNAMIC 3D EVALUATION

**BTimer (GEN3C) comparison.** For dynamic 3D evaluation, we compare our method with BTimer (GEN3C), as discussed in Sec. D.1, on 100 out-of-distribution videos from our dynamic *Lyra* dataset. We crop the video to $512 \times 512$ for fair comparisons, as BTimer was mainly trained on that resolution. We show results in Tab. 6b and observe that our method significantly outperforms BTimer (GEN3C).

### D.4 DEPTH VISUALIZATION

We visualize depths from our generated 3DGS in Fig. 11. Using depth supervision prevents flat geometries without sacrificing visual quality.

Ours          Ours (w/o depth loss)

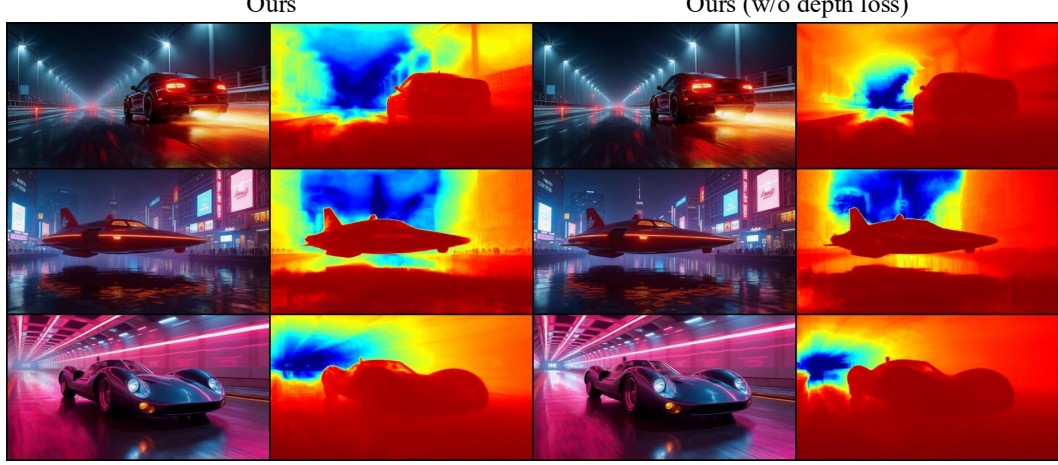

Figure 11: **Depth loss ablation.** We visualize 3DGS renderings and corresponding depth maps. Using the depth loss as additional supervision prevents flat geometries without sacrificing visual quality.

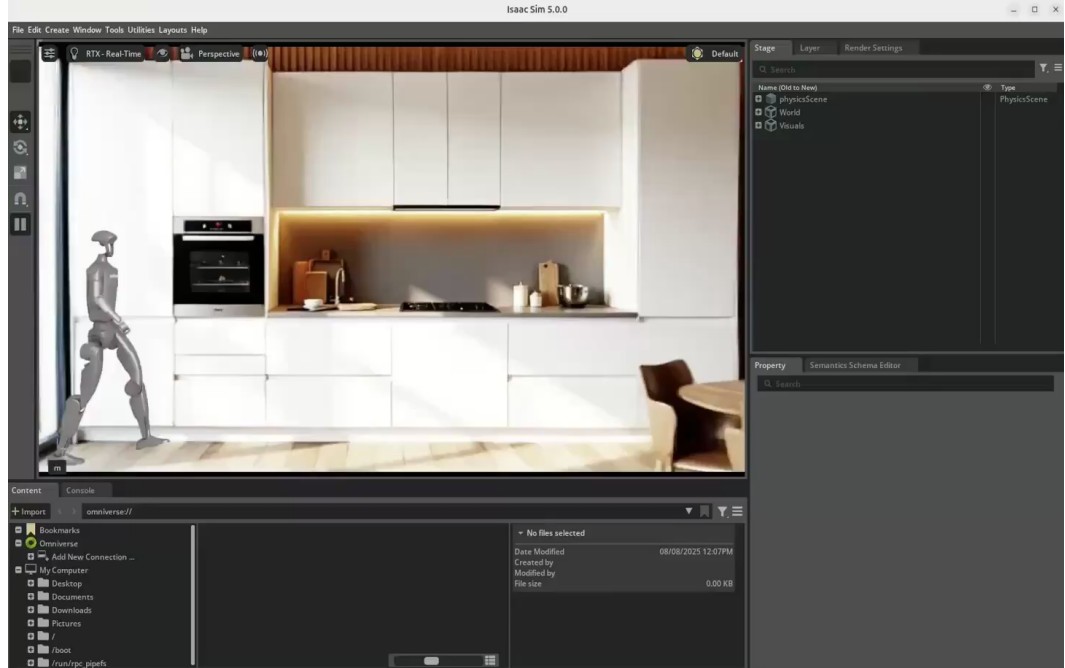

Figure 12: **Robot Simulation.** We visualize a frame of a robot simulation within the Isaac Sim 5.0 simulation framework that takes a generated 3DGS scene from our method as input.

## E   ROBOT SIMULATION

One of our primary motivations for building a diverse 3D generator is its ability to synthesize simulation environments for autonomous agents. To this end, we develop an end-to-end pipeline: we first generate 3D scenes from text with our model, export the 3D Gaussians as a .ply file, and convert the .ply files into a .usdz format. We use 3DGUT (Wu et al., 2025a) and its export function to create .usdz files. The resulting .usdz file can then be imported into the NVIDIA Isaac robot development platform as a physically based virtual environment. AI-based robots can be trained and tested under diverse conditions within these environments. We present early demo results on our supplementary webpage and a screenshot of the interface in Fig. 12.

## F   RELATED WORK

In this section, we will discuss additional related work. We highlight the differences to previous works CAT3D (Gao et al., 2024b) and Bolt3D (Szymanowicz et al., 2025b) in Fig. 13.

**3D generation.** Early approaches to 3D generation primarily focused on single object categories, extending GANs into the 3D domain by leveraging neural renderers as an inductive bias (DeVries et al., 2021; Chan et al., 2022; Or-El et al., 2022; Schwarz et al., 2022; Bahmani et al., 2023b). With the introduction of CLIP-based supervision (Radford et al., 2021), the field advanced toward more flexible and diverse asset creation, enabling both text-driven generation and editing (Chen et al., 2018; Jain et al., 2022; Sanghi et al., 2022; Jetchev, 2021; Gao et al., 2023; Wang et al., 2022). More recently, diffusion models have significantly improved quality by replacing CLIP guidance with Score Distillation Sampling (SDS) (Poole et al., 2023; Wang et al., 2023b; Lin et al., 2023a; Chen et al., 2023b; Liang et al., 2023; Wang et al., 2023a; Li et al., 2024e; He et al., 2024b; Ye et al., 2024; Liu et al., 2024b; Yu et al., 2023; Katzir et al., 2024; Lee et al., 2024a; Sun et al., 2024a).

To further enhance structural coherence, several methods enforce multi-view consistency by generating scenes from different perspectives (Lin et al., 2023b; Liu et al., 2023; Shi et al., 2024; Feng et al., 2024a; Liu et al., 2024a; Kim et al., 2023; Voleti et al., 2024; Höllein et al., 2024; Tang et al., 2024c; Gao et al., 2024b; Wang et al., 2025f; Kant et al., 2025; Yuan et al., 2025; Ren et al., 2024b), while others adopt iterative inpainting as a strategy for scene synthesis (Höllein et al., 2023; Shriram et al., 2024). Another line of work explores lifting 2D images into 3D using NeRF (Mildenhall et al., 2020),

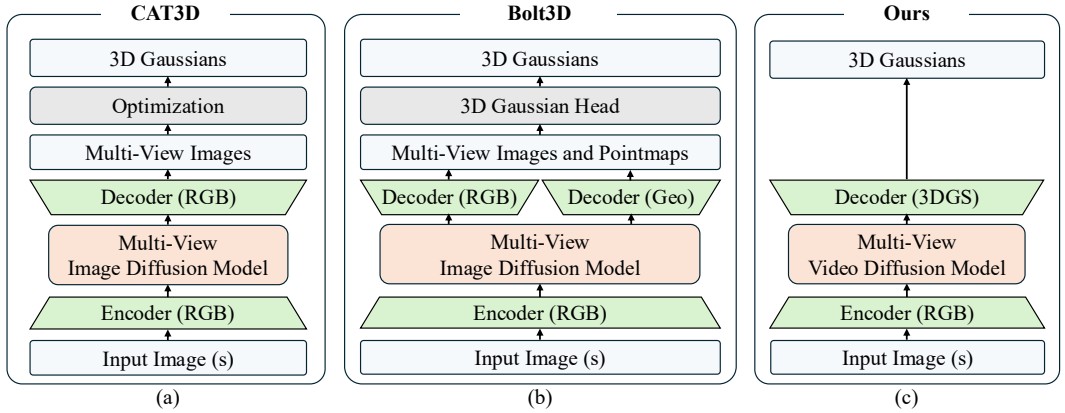

Figure 13: **Approaches for 3D Generation.** CAT3D (Gao et al., 2024b) proposed a multi-view image diffusion model that outputs images from novel viewpoints; subsequently, the images are reconstructed into 3D Gaussians using optimization. Bolt3D (Szymanowicz et al., 2025b) fine-tunes CAT3D to output pointmaps using a geometry autoencoder; instead of relying on optimization, the 3D scene is predicted with a feed-forward 3D Gaussian head. In contrast, our work builds upon a pre-trained camera-controlled video diffusion model and *directly* decodes the multi-view video latents into 3D Gaussians.

3D Gaussian Splatting (Kerbl et al., 2023), or mesh-based representations coupled with diffusion models (Chan et al., 2023; Tang et al., 2023a; Gu et al., 2023; Liu et al., 2024c; Yoo et al., 2023; Tewari et al., 2023; Qian et al., 2024b; Long et al., 2024; Wan et al., 2024; Szymanowicz et al., 2023; Lu et al., 2024).

**Feed-forward 3D models.** Complementary efforts investigate fast, feed-forward techniques that directly predict 3D content from images or text inputs (Hong et al., 2024; Li et al., 2024b; Xu et al., 2024e;d; Zhang et al., 2024a; Han et al., 2024; Jiang & Wang, 2024; Xie et al., 2024a; Tang et al., 2024b; Tochilkin et al., 2024; Qian et al., 2024a; Szymanowicz et al., 2024; 2025a; Liang et al., 2025a; Szymanowicz et al., 2025b; Schwarz et al., 2025; Yang et al., 2025; Zhang et al., 2025a). However, in contrast to our method, these approaches remain limited to producing static 3D scenes. Other methods focus on specific scenes such as faces (Kirschstein et al., 2025). Concurrent work (Liang et al., 2025b; Xu et al., 2025) tackles real scenes but can not handle diverse generated scenes or large viewpoint changes.

**4D generation.** Recent years have seen rapid progress in 4D generation, i.e., dynamic 3D scene synthesis. Many approaches leverage input text prompts or images as guidance. Following the introduction of large-scale generative models for this task (Singer et al., 2023), subsequent works have made notable advances in both visual fidelity and motion quality (Ren et al., 2023; Ling et al., 2024a; Bahmani et al., 2024b; Zheng et al., 2024b; Gao et al., 2024a; Yang et al., 2024a; Jiang et al., 2024b; Miao et al., 2024; Li et al., 2024a; Zhang et al., 2024g; Yuan et al., 2024; Jiang et al., 2024a).

While text conditioning is widely used, alternative methods aim to lift 2D images or videos into dynamic 3D scenes (Ren et al., 2023; Zhao et al., 2023; Yin et al., 2023; Pan et al., 2024; Zheng et al., 2024b; Ling et al., 2024a; Gao et al., 2024a; Zeng et al., 2024b; Chu et al., 2024; Wu et al., 2024; Yang et al., 2024b; Wang et al., 2024c; Feng et al., 2024b; Sun et al., 2024c; Zhang et al., 2024c; Ren et al., 2024a; Lee et al., 2024b; Li et al., 2024c; Van Hoorick et al., 2024; Uzolas et al., 2024; Chai et al., 2024; Liang et al., 2024; Li et al., 2024d; Xie et al., 2024b; Li et al., 2025a; Wang et al., 2025e; Zhu et al., 2025; Zhou et al., 2025; Hu et al., 2025; Park et al., 2025). Other efforts explore incorporating physics priors into generation pipelines (Lin et al., 2024; Huang et al., 2024b; Zhang et al., 2024f; Kiray et al., 2025), or enhancing motion controllability with template-driven approaches (Zhang et al., 2024d; Sun et al., 2024b; Chen et al., 2024b). A parallel direction emphasizes compositional and interactive 4D generation (Bahmani et al., 2024a; Xu et al., 2024b; Cao et al., 2024; Yu et al., 2024a; Zeng et al., 2024a; Zhu et al., 2024).

Another strand of research extends 3D GAN frameworks into the 4D setting by training directly on 2D video data (Bahmani et al., 2023a; Xu et al., 2023). However, such methods are typically limited by small-scale, single-category datasets, which constrain generalization. Moreover, most existing

approaches remain object-centric, often neglecting background elements. As a result, their overall visual fidelity lags behind the high photorealism achieved by recent video generation models, which our method leverages.

**Camera-conditioned video models.** Recent progress has focused on incorporating camera control into video diffusion models. The pioneering work MotionCtrl (Wang et al., 2024e) introduced camera conditioning by augmenting pre-trained video models (Chen et al., 2023a; Blattmann et al., 2023) with extrinsic matrices. Subsequent efforts (He et al., 2024a; Xu et al., 2024c; Kuang et al., 2024; Ju et al., 2025; He et al., 2025; Li et al., 2025c; Wang et al., 2025c; 2024d; Lei et al., 2025; Liu et al., 2025) enhanced conditioning by representing cameras using Plücker coordinates, while others use follow MotionCtrl to simply use poses flattened (Wang et al., 2025b). Another line of research (Hu et al., 2024; Xiao et al., 2024; Ling et al., 2024c; Hou et al., 2024) enabled camera motion control without introducing additional trainable parameters. Despite these advances, all of the above rely on U-Net-based architectures. Other approaches use low-rank approaches for camera control (Zhang et al., 2025b).

Several recent works continue to extend U-Net-based approaches for camera control (Zhao et al., 2024; Xu et al., 2024a; Zheng et al., 2024a; Zhang et al., 2024b; Yu et al., 2024b; Lei et al., 2024), while others begin exploring diffusion transformers. Cheong et al.(Cheong et al., 2024) propose one such transformer-based framework, though its scene and visual quality remain limited. DimensionX(Sun et al., 2024d) introduces joint space-time control in diffusion transformers, but relies on pre-defined, non-continuous camera trajectories. Other works (Huang et al., 2024a; Mai et al., 2025; Jiang et al., 2025b) instead explore pose and geometry estimation with a video DiT. CAT4D (Wu et al., 2025b) proposes a multi-view video diffusion model derived from fine-tuning a multi-view backbone. Large-scale synthetic data generation further boosts performance, as demonstrated by SynCamMaster (Bai et al., 2025b) and ReCamMaster (Bai et al., 2025a), which achieve strong results on camera-controlled video generation using transformer-based architectures. Finally, 4DiM (Watson et al., 2024) trains a space-time diffusion model from scratch for novel view synthesis from a single input image. Another line of work edits camera movements of videos with underlying motion using camera retargeting (Seo et al., 2025; Ma et al., 2025). One popular direction has been to use a spatio-temporal cache, e.g., using point clouds or optical flow, to condition the camera-controlled video generation (Wu et al., 2025c; Ren et al., 2025; Wang et al., 2025d; Cao et al., 2025; YU et al., 2025; Jin et al., 2025; Xing et al., 2025; Yan et al., 2025; Li et al., 2025b; Gu et al., 2025).

