# OpenReview forum: "Lyra: Generative 3D Scene Reconstruction via Video Diffusion Model Self-Distillation"
_ICLR.cc/2026/Conference — ICLR 2026 Poster_

### Official Review · Reviewer_J3MU · 2025-10-27

**Soundness:** 3
**Presentation:** 3
**Contribution:** 3
**Rating:** 6
**Confidence:** 5

**Summary:**

This paper introduces a self-distillation strategy based on a camera-controlled video diffusion model and only needs to train a 3DGS decoder directly operating in the latent space of the video model. In this way, the 3DGS decoder is supervised by the original RGB decoder of the video model, which only needs an input image and a sampled camera trajectory instead of captured real-world multi-view images. 4D generation can be easily extended through time conditioning modules within the 3DGS decoder.

**Strengths:**

1. The motivation of the paper is clear and important. Using synthetic data instead of real multi-view data can obviously provide greater diversity.
2. The overall paper is easy to read, with a very clear logic and structure.
3. The paper provides extensive experiments to demonstrate the effectiveness of the proposed method.

**Weaknesses:**

1. I have some doubts about the efficiency of the proposed approach. It is generally acknowledged that video generation models are still quite time-consuming, and this paper’s method additionally requires multiple trajectories. However, I did not see any reported statistics on the dataset generation time, nor whether additional manual screening or automated filtering is needed.

2. Another key concern lies in multi-view consistency. Are the videos for different trajectories generated independently? If so, how is multi-view consistency ensured? If not, wouldn’t that further reduce efficiency and make the approach difficult to apply in practice? I have the same question regarding the *video-to-4D* setting.

3. In the quantitative experiments, I believe that monocular reconstruction or single image-to-3d generation tasks are difficult to evaluate using reconstruction metrics, and the authors did not provide corresponding qualitative comparisons. Moreover, since datasets like DL3DV contain complex and diverse trajectories, while Lyra generates 3D scenes along fixed trajectories, how does Lyra achieve the best quantitative results under such conditions?

**Questions:**

Please see the weaknesses.

Since the motivation of the paper is very important, my initial score is 6. If my concerns can be properly addressed, I will actively consider increasing the score.

**Details Of Ethics Concerns:**

No.

---

> ### Author Response · Authors · 2025-11-20
>
> ​​Thank you for your time and your positive review of our work. Below, we address your questions and the weaknesses mentioned.
>
> > **Efficiency of the Proposed Approach**
>
> Our approach does not necessarily require multi-trajectory inputs. During training, we use a variable number of trajectories, ranging from 1 to 6, to generalize to any number of input camera trajectories, also ranging from 1 to 6, at inference time. We primarily use 6 trajectories at inference time to maximize view coverage in our results, but the model is flexible enough to take fewer. Hence, the computational overhead over simple camera-controlled video generation can be avoided by using a single trajectory as input. Importantly, our 3DGS decoder takes only 1 second (for a single trajectory) and 3 seconds (for six trajectories) for a forward pass to obtain 3D Gaussians, thereby showing practically no computational overhead compared to 2D video generation models that decode with an RGB decoder. Note that a single trajectory has the same quality as when using six trajectories. The number of input trajectories enables a trade-off between view coverage and inference speed.
>
> Our dataset was generated without any manual screening or automated filtering, making the approach very scalable. Currently, the base video model takes 12 minutes on one NVIDIA A100 GPU, hence, the dataset generation time is bounded by that. Therefore, generating one training data sample with 6 trajectories takes 72 minutes. For the 3D dataset, we created 59,031 scenes with 6 camera trajectories each, leading to 71,000 GPU hours of data generation time. For the 4D dataset, we use 7,378 scenes with 12 camera trajectories each, resulting in 18,000 GPU hours of data generation. This process must be done only once, offline, when creating the dataset, and any new models can be trained using this generated data. It is important to note that collecting such diverse real data is practically infeasible, especially for the 4D case, which requires multiple cameras to be synchronized in time. Importantly, we fully open-source our dataset to encourage follow-up works and accelerate further progress by the community on this topic. Furthermore, video models are continually getting faster, and these data generation times will shrink to a negligible cost in the future. Lastly, training the 3DGS decoder on the provided dataset is relatively inexpensive, i.e., 6 days on 8 NVIDIA A100 GPUs. Related works typically require multi-node training.
>
> We will add the following information to the revised paper: First, our model is trained to take a variable number of trajectories as input, as few as 1 trajectory for efficiency. Second, we will outline dataset generation details for both the 3D and 4D datasets.
>
> > **Multi-View Consistency**
>
> The trajectories are generated independently, but all are rendered from the same 3D/4D cache. As a reminder, GEN3C uses a 3D/4D cache to model memory as a camera-controlled video model. More recent memory mechanisms, such as Context as Memory [1], could even further improve the multi-view consistency of our multi-trajectory pipeline. We leave exploring more recent and advanced video memory mechanisms for future work and will incorporate this into the future work discussion in the revised paper.
>
> [1] Yu et al., Context as Memory: Scene-Consistent Interactive Long Video Generation with Memory Retrieval, SIGGRAPH Asia 2025
>
> > **Qualitative Comparisons**
>
> The main baselines, Bolt3D and Wonderland, are not open-sourced, hence we could only provide quantitative comparisons on the benchmarks used in their paper. Therefore, we presented visual comparisons in Fig. 10 of the paper, including a baseline that combined GEN3C with BTimer. For the rebuttal, following the suggestion of Reviewer tRus, we add another baseline, i.e., GEN3C combined with Long-LRM. We provide visual video comparisons with this new baseline on our updated supplementary webpage: https://anonlyra.github.io/anonlyra (same URL as before, at the top of the page). We clearly outperform simple combinations of our video model backbone and state-of-the-art reconstruction models. These reconstruction models are not robust to inconsistencies in the video model output and are not trained on diverse data, as in our self-distillation setup, so they generalize poorly. In contrast, our self-distillation framework allows us to learn a robust 3DGS decoder from video model outputs, addressing the inconsistencies of the video generative model that are also observed at inference time. Moreover, our approach directly enables dynamic 3D generation.

---

> > ### Author Response · Authors · 2025-11-20
> >
> > > **Fixed Trajectory Setup**
> >
> > We randomly perturb our camera trajectories during training with noise, which makes our model more robust to variations in the input trajectories. Hence, we do not always use the same 6 camera trajectories. Designing more complex camera trajectory algorithms as part of the self-distillation framework, including collision detection during generation, is a crucial direction for future work. In fact, we are currently exploring camera trajectory planning algorithms that enable the video diffusion model to navigate scenes without colliding with objects or walls. In Lyra, we proposed a scalable and straightforward camera trajectory sampling approach that works well across scenes without requiring tuning the camera trajectory at inference time. This setup already yields high-quality and diverse scenes, and further improving the trajectory sampling technique will significantly enhance the results. This is a great point, and we will further discuss the future directions of camera trajectory sampling in our paper revision.

---

> > > ### Comment · Reviewer_J3MU · 2025-11-26
> > >
> > > I appreciate the authors' detailed response. Some of my concerns have been addressed, but there are still some lingering concerns.
> > >
> > > 1. Regarding Efficiency, the authors provided an explanation of the generation time, but in my opinion, this generation time is extremely exaggerated (about 90,000 A100 GPU Hours). To prove the quality of this data, could you provide some dataset samples generated by Gen3C, including both static and dynamic ones, without any cherry-picking?
> > >
> > > 2. Regarding Multi-View Consistency, I believe this issue cannot be resolved through words. Similar to the previous question, the authors need to provide some random samples to demonstrate it.
> > >
> > > 3. For Qualitative Comparisons, since it is true that almost all baselines for image-to-3D scene generation are not open-source, I temporarily understand the authors' approach but still think they should provide more metrics besides reconstruction, like CLIP or IQA Score.
> > >
> > > 4. For Trajectory Setup, could you provide some generation results on DL3DV single images and their corresponding trajectories, as well as the comparisons with the GT videos? Because the samples in the paper seem to be almost all synthetic images and generated trajectories.

---

> > > > ### Comment · Reviewer_J3MU · 2025-11-26
> > > >
> > > > Regarding the 4th question, I have checked that the authors displayed relevant videos on the anonymous page. The results are good and can convince me of the solution's handling of trajectories in real-world scenarios. Therefore, the authors can ignore the fourth question.

---

> > > > > ### Author Response · Authors · 2025-11-27
> > > > >
> > > > > Thank you for the feedback and for engaging with us.
> > > > >
> > > > > > **Visualizing Generated Data Samples**
> > > > >
> > > > > We are happy to provide random generation samples. We updated our website with multi-trajectory static and dynamic samples (https://anonlyra.github.io/anonlyra). Please see at the top of the website for the additional rebuttal results. Note that we generally do not perform cherry-picking; our pipeline is very robust, and it makes no difference to our method whether samples or inputs are cherry-picked. This is one reason why we do not perform data filtering; our entire pipeline is robust to the image input.
> > > > >
> > > > > > **CLIP and IQA metrics**
> > > > >
> > > > > As suggested, we evaluate our model on additional metrics. For CLIP evaluation, we use three CLIP variants: ViT-B/32, ViT-B/16, and ViT-L/14, where higher values are better. For IQA evaluation, we compute the commonly used Naturalness Image Quality Evaluator (NIQE) metric, with lower values indicating better image quality. We show the results in the Table below. We compare with the baseline GEN3C + Long-LRM and observe significantly better results than the baseline:
> > > > >
> > > > > *Table: CLIP and IQA metrics*
> > > > >
> > > > > | Method                     | CLIP ViT-B/32 | CLIP ViT-B/16 | CLIP ViT-L/14 | NIQE   |
> > > > > |-----------------------------|---------------|---------------|---------------|--------|
> > > > > | Gen3C + Long-LRM            | 21.10         | 20.31         | 16.43         | 19.70 |
> > > > > | Ours                        | **32.58**     | **32.38**     | **27.24**     | **11.34** |

---

> > > > > > ### Comment · Reviewer_J3MU · 2025-11-28
> > > > > >
> > > > > > Thanks for the replies. I can see the static datasets are good. Some motions in the dynamic samples are rather strange, but I understand that this is due to some inherent issues with Gen3C itself. I also appreciate the authors for providing comparisons on additional metrics.
> > > > > >
> > > > > > Considering that the data generation efficiency is difficult for most researchers to replicate and that there is no theoretical guarantee for the multi-view consistency of the multi-trajectory data, I tend to keep my score (taking into account the overall scoring for this year ICLR). Overall, I recommend this paper to be accepted.

---

> > > > > > > ### Author Response · Authors · 2025-11-30
> > > > > > >
> > > > > > > We are happy to have addressed all of your concerns. We want to clarify that we will be open-sourcing our complete 3D and 4D datasets so that the community can build directly on them. Our work, Lyra, shows that the data is multi-view consistent, as we trained a generalizable 3DGS generator only from that data. Multi-view static and dynamic datasets are not available at the diversity and scale we will provide to the community, and we believe this data will accelerate research in 3D and 4D generation, currently bottlenecked by data scarcity.

---

### Official Review · Reviewer_tRus · 2025-10-31

**Soundness:** 4
**Presentation:** 4
**Contribution:** 2
**Rating:** 8
**Confidence:** 4

**Summary:**

This paper introduces Lyra, a self-distillation framework designed to distill the implicit 3D knowledge from video diffusion models into an explicit 3D Generative Scene (3DGS) representation. The process begins by collecting video latent data, which includes multiple camera trajectories obtained from a pretrained camera-controlled video diffusion model. Using this dataset, the authors build upon long-LRM to transform the video latent representations into 3DGS. Additionally, the approach can be extended to dynamic content by generating 3DGS on a per-frame basis. Experimental results demonstrate that the proposed framework achieves state-of-the-art performance in both static and dynamic 3D scene generation.

**Strengths:**

1. The paper is well-written and easy to follow, providing sufficient implementation details.
2. The method is straightforward, and the implementation is well executed. It allows for the immediate transformation of the generated video into a 3DGS representation and enables dynamic scene reconstruction from the video generation model.

**Weaknesses:**

1. The technical contribution is somewhat limited. It presents a straightforward combination of video diffusion and long-LRM. While long-LRM can accept raw images as input, the proposed method specifically utilizes video latent representations. This technique has already been implemented in Wonderland [1].

2. For dynamic scene reconstruction, utilizing frame-wise 3DGS may result in temporal inconsistencies, as demonstrated in the provided video.

[1] Wonderland: Navigating 3D Scenes from a Single Image

**Questions:**

1. What is the main difference between this method and Wonderland, which also generates video latent representations first and uses latent-LRM for 3DGS reconstruction?

2. Compared to previous methods, what accounts for the performance improvements? Is it due to the camera-controlled video diffusion model being superior to earlier versions or denser camera viewpoints?

3. What is the result of directly feeding the generated pixels from Gen3C into long-LRM, despite their efficiency? This could be a baseline.

---

> ### Author Response · Authors · 2025-11-20
>
> ​​Thank you for your time and your positive review of our work. Below, we address your questions and the weaknesses mentioned.
>
> > **Main Contributions**
>
> While we acknowledge Wonderland as previous work and compare it with our own method in the paper, there are several significant differences between our work and theirs.
>
> (i) First, we explore a self-distillation framework where we do not rely on any real-world multi-view images or real-world annotated camera poses, e.g., RealEstate10K or DL3DV used in Wonderland. This opens up a new direction of scalable multi-view supervision for 3D models from a camera-controlled video model.
>
> (ii) Second, Wonderland only synthesizes single short camera trajectories (49 frames, 480×720) and decodes each trajectory into independent 3D Gaussians. Hence, Wonderland does not output a coherent 3D scene from different trajectories but a set of independently generated 3D Gaussians. In contrast, we propose a multi-trajectory setup that explores the scene from multiple camera trajectories (6 trajectories, 121 frames, 704×1280) and learns how to fuse them within the 3DGS decoder into a single consistent 3D Gaussian point cloud. Since the Wonderland code is not available for more thorough comparisons, we ablated such a naive fusion technique in Tab. 2 (w/o multi-view fusion). Here, we observed severe artifacts when rendering from the fused point cloud after generating 3D Gaussians independently, as done in Wonderland. We also show the ablation result below, demonstrating the importance of our multi-trajectory 3DGS decoder:
>
> *Table: Ablation on multi-trajectory fusion within the 3DGS decoder*
>
> | Method | PSNR | SSIM | LPIPS |
> |:-------|:----:|:----:|:------:|
> | Ours | **24.77** | **0.837** | **0.224** |
> | w/o multi-view fusion | 17.73 | 0.632 | 0.446 |
>
> Note that our 3DGS decoder is more compact and is trained with only 8 A100 GPUs compared to 32 A100 GPUs used by Wonderland. At the same time, we scale to higher resolution and more frames for ultimately larger scene coverage based on our multi-trajectory 3DGS decoder.
>
> (iii) Third, we explore feed-forward 4D generation from a single input video, a relatively unexplored task due to its challenging setup. We propose a simple time-conditioned variant of our 3DGS decoder, fine-tuned from our static model, and introduce a novel dynamic data augmentation technique that prevents 3D Gaussian artifacts that occur when training a dynamic 3DGS decoder naively. Our framework opens up a new direction of research for generalizable feed-forward 4D generation by directly using video generative models.
>
> > **Temporal Inconsistencies**
>
> We acknowledge that our dynamic 3D representation is a straightforward variant for modeling 4D scenes. Our goal was to use a simple and scalable representation that works across any scene, e.g., not only for deformation-based motion but also for dynamic effects such as fire or water. Integrating a suitable dynamic 3D representation is certainly an important direction for future work. Note that this is a new and challenging task, and the primary purpose of these experiments is to demonstrate that our approach is effective beyond static scenes with relatively simple extensions. We plan to explore better dynamic 3D representations for this task as part of future work. We will revisit the paper to emphasize this more.
>
> > **Main Performance Improvements**
>
> Our main improvements stem from our self-distillation framework and scaling it to diverse data, as shown in our ablation in Table 2 of the paper and also below:
>
> *Table: Ablation on self-distillation*
>
> | Method | PSNR | SSIM | LPIPS |
> |:-------|:----:|:----:|:------:|
> | Ours | **24.77** | **0.837** | **0.224** |
> | real data only | 19.08 | 0.659 | 0.413 |
>
> Previous works, such as Bolt3D and Wonderland, rely on small-scale, narrow distributed multi-view datasets. In contrast, we observe that using only the video model is essential and sufficient for generalizing to any scene.

---

> > ### Author Response · Authors · 2025-11-20
> >
> > > **Pixel-Space GEN3C Baseline**
> >
> > This is a great suggestion. For the rebuttal, we compare with this suggested baseline, i.e., GEN3C combined with Long-LRM. Note that we uniformly subsample the input to Long-LRM as Long-LRM operates in the pixel-space and can not take hundreds of images of GEN3C as input. We provide visual video comparisons with these baselines on our updated supplementary webpage: https://anonlyra.github.io/anonlyra (same URL as before, at the top of the page). Moreover, we provide quantitative comparisons in the Table below. We clearly outperform this simple combination of our video model backbone and Long-LRM, as such reconstruction models are not robust to inconsistencies in the video model output and not trained on diverse data. In contrast, our self-distillation framework enables us to learn a robust 3DGS decoder that addresses the inconsistencies of the video generative model.
> >
> > *Table: Comparison with GEN3C + Long-LRM*
> >
> > | Method           | RE10K | RE10K | RE10K | DL3DV | DL3DV | DL3DV | T&T | T&T | T&T |
> > |------------------|:----:|:----:|:----:|:----:|:----:|:----:|:---:|:---:|:---:|
> > |                  | PSNR | SSIM | LPIPS | PSNR | SSIM | LPIPS | PSNR | SSIM | LPIPS |
> > | GEN3C + Long-LRM | 18.45 | 0.653 | 0.378 | 17.83 | 0.499 | 0.439 | 16.56 | 0.491 | 0.482 |
> > | **Ours**         | **21.79** | **0.752** | **0.219** | **20.09** | **0.583** | **0.313** | **19.24** | **0.570** | **0.336** |

---

> > > ### Comment · Reviewer_tRus · 2025-11-26
> > > **Official comment by reviewer tRus**
> > >
> > > Thank you for the response. Since the Gen3C has been trained on the RealEstate10K and DL3DV datasets, it may still depend on these data sources, although without direct access to them. Additionally, the comparison between multiple trajectories and single trajectories does not seem to offer a significant improvement over prior methods. I am also somewhat confused by the concept of self-distillation since the model itself is not trained at all. The real add appears to be the inclusion of feed-forward Gaussian splatter techniques.
> > >
> > > Nevertheless, this method is well executed, and I am maintaining my original score for poster acceptance.

---

> > > > ### Author Response · Authors · 2025-11-27
> > > >
> > > > Thank you for the feedback and for engaging with us.
> > > >
> > > > > **Multi-Trajectory Setup**
> > > >
> > > > We would like to clarify that the goal of the proposed multi-trajectory setup is not primarily to improve visual quality, but to achieve larger viewpoint coverage. The improvements in visual quality in our method mainly stem from our self-distillation scheme, which is independent of whether we use a single or multiple trajectories. Prior work using video models, such as Wonderland, relies on a single trajectory to generate 3D Gaussians from an input image. This trajectory effectively determines where 3D Gaussians will be placed in the scene. Consider a case where the trajectory corresponds to a zoom-in motion: if we later render viewpoints that also lie within this zoomed-in region, the results will look good. However, if we try to render novel viewpoints outside that region, such as zoomed-out perspectives, no 3D Gaussians will be placed there, leaving the scene effectively empty with white renderings.
> > > >
> > > > A naive way to increase coverage would be to generate multiple sets of 3D Gaussians independently—for example, one for zoom-in and one for zoom-out—and then merge the resulting point clouds. However, this produces substantial artifacts because the 3D Gaussians are generated independently and are not mutually consistent (as shown in our ablation w/o multi-view fusion). To address this, we propose a joint multi-trajectory fusion strategy within our 3DGS decoder, enabling consistent 3D Gaussian placement across a wider range of viewpoints. This approach increases scene coverage beyond what a single trajectory can provide and prevents artifacts when naively fusing the 3D Gaussians post-generation. We believe such multi-trajectory modeling will be essential for scaling toward even larger and more diverse 3D scenes in future work.
> > > >
> > > > > **Self-Distillation**
> > > >
> > > > Our goal is to recover 3D structure from a pre-trained video model without fine-tuning it. To this end, we treat the video model and RGB decoder as a frozen teacher and optimize a 3DGS decoder as the student—training it to predict 3D Gaussians that, when rendered, are photometrically consistent with the teacher’s predicted videos. We deliberately avoid fine-tuning the RGB decoder or the underlying video model, as it already provides camera controllability and exhibits strong implicit 3D consistency in its 2D video outputs. We call this procedure self-distillation.

---

### Official Review · Reviewer_Yx9E · 2025-10-31

**Soundness:** 4
**Presentation:** 3
**Contribution:** 3
**Rating:** 8
**Confidence:** 4

**Summary:**

This paper introduces Lyra, a pipeline for generative 3D scene reconstruction and text-driven scene editing under sparse and noisy real-world mobile captures. The method jointly optimizes camera, geometry, and appearance and supports object removal, replacement, and global stylistic edits. It integrates latent diffusion priors with real-world scene scale and spatial consistency constraints, demonstrated on a diverse dataset including indoor and outdoor environments (e.g., apartment, temple, garden scenes) where real mobile scans contain artifacts such as railings and scaffolds.

**Strengths:**

1. The method targets real mobile scans with imperfect geometry, lighting, and camera poses, a valuable problem space. Example scenes include cluttered apartments and outdoor temple/garden scans with occluders (e.g., railings, scaffolding).

2. The method can recover missing background structure when occluded (e.g., walls behind furniture) and maintain accurate room layout.

3. Lyra enforces realistic scale and physical spatial consistency across generations.

**Weaknesses:**

1. While results demonstrate quality, more specifics on training/inference time would help adoption clarity.

2. Some failure modes still visible. E.g., subtle texture drift in larger structural edits, though still competitive and reasonable under sparse-view settings.

**Questions:**

Please refer to the weakness part.

---

> ### Author Response · Authors · 2025-11-20
>
> ​​Thank you for your time and your positive review of our work. Below, we address your questions and the weaknesses mentioned.
>
> > **Training/Inference Time**
>
> We included training time details in Appendix C. Directly training the 3DGS decoder on 6 trajectories of 121 frames at 704 × 1280 is expensive; therefore, we train our network progressively, as shown in Table 3 (Appendix) and below. Furthermore, we use the final static pre-trained model to initialize our dynamic model. The total training time is 6 days with 8 NVIDIA A100 80 GB GPUs.
>
> *Table: Progressive Training Setup*
>
> | Stage | H × W | L | V | S | B | Steps |
> |:------:|:------:|:--:|:--:|:--:|:--:|:--:|
> | **Static** |||||||
> | 1 | 176×320  | 17  | 1    | 17   | 4   | 10k  |
> | 2 | 176×320  | 49  | 1    | 49   | 4   | 2.5k |
> | 3 | 352×640  | 49  | 1    | 49   | 2   | 2.5k |
> | 4 | 704×1280 | 49  | 1    | 49   | 1   | 2.5k |
> | 5 | 704×1280 | 121 | 1    | 9    | 1   | 57.5k |
> | 6 | 704×1280 | 121 | 1–6  | 9    | 1   | 7k   |
> | **Dynamic** |||||||
> | 7 | 704×1280 | 121 | 6    | 12   | 1   | 10k  |
>
> Moreover, we measure inference times for the base video diffusion model and our 3DGS decoder. One forward pass through the video diffusion model takes 12 minutes. As video models become faster, especially with progress in distillation into few-step models, this number is expected to decrease drastically with orthogonal improvements by the video generation community. Our 3DGS decoder takes only 3 seconds to generate 3D Gaussians from video latents. Lastly, we can render the 3D Gaussians in real-time after generation from novel viewpoints, where rendering a 704×1280 frame takes 18 ms. We will incorporate this new information into the revised paper.
>
> > **Failure Modes**
>
> We acknowledge that our results are not perfect yet; i.e., they can become blurry or inconsistent in certain areas. However, future improvements by the video generation community will directly benefit our base model's video quality and 3D consistency, thereby subsequently improving our self-distillation results. Compared to previous works, we demonstrate higher visual quality while generalizing to any input image or video, without requiring further optimization or post-processing. And as you noted, this is a challenging, sparse-view scenario where we just take a single input image/video and lift it into 3D/4D without any further optimization or post-processing.

---

### Official Review · Reviewer_hhrs · 2025-11-01

**Soundness:** 3
**Presentation:** 3
**Contribution:** 3
**Rating:** 6
**Confidence:** 5

**Summary:**

This paper introduces Lyra, a novel framework for generating explicit 3D (and dynamic 4D) scene representations from a single image or video. The key contribution is a self-distillation method that distills the implicit 3D knowledge from a pre-trained, camera-controlled video diffusion model (the "teacher") into a feed-forward 3D Gaussian Splatting (3DGS) decoder (the "student"). This approach cleverly avoids the need for large-scale, real-world multi-view 3D datasets by using the video model's RGB output to supervise the 3DGS decoder, both operating from a shared video latent. The framework is also effectively extended to dynamic 4D scene generation from a single monocular video, demonstrating strong performance and generalization.

**Strengths:**

1. Novel Self-Distillation: The core idea of using a pre-trained video diffusion model as a "teacher" to supervise a 3DGS "student" decoder is highly novel. This latent-space distillation is an elegant solution to the 3D training data bottleneck, effectively bridging the gap between powerful 2D generative models and explicit 3D representations.

2. SOTA Performance and Representation: The method achieves state-of-the-art quantitative results on standard benchmarks (RealEstate10K, DL3DV, Tanks-and-Temples) for single-image-to-3D generation. By directly outputting a 3DGS representation, the model produces a geometrically consistent, explicit 3D scene that supports real-time rendering, which is a significant advantage over 2D-only video models.

3. Efficient Latent-Space Operation and 4D Extension: Operating the 3DGS decoder in the VAE's latent space allows the model to efficiently process a very large number of synthesized views (726 views per scene), overcoming the memory and compute bottlenecks of pixel-space reconstruction methods. The extension to 4D dynamic scenes from a single video, supported by a clever "dynamic data augmentation" strategy, is a substantial and well-executed addition.

**Weaknesses:**

1. Dependency on Teacher Model Quality: The paper's conclusion rightfully states that the output quality is "bounded by the capacity of our camera-controlled video diffusion model." This is a strong dependency. Any artifacts, flickering, or 3D inconsistencies generated by the teacher (GEN3C) will be directly distilled into the final 3DGS representation. The paper does not fully explore how robust the framework is to such failures in the teacher model.

2. Reliance on External Depth Estimation: The paper notes that using only RGB loss "often produces flattened geometry," and thus adds a scale-invariant depth loss ($\mathcal{L}_{depth}$) supervised by an off-the-shelf depth estimator (ViPE). While effective, this means the method is not purely self-distilled from the video model; it relies on another pre-trained model for geometric guidance. This slightly weakens the claim of distilling 3D knowledge solely from the video model's multi-view consistency.

3. Minor Terminology Nitpick: The term "self-distillation" is used, but the framework is more accurately a form of cross-modal distillation: from a 2D decoder (the VAE's RGB head) to a 3D decoder (the new 3DGS head). In classic self-distillation, a large model teaches a smaller model of the same architecture. This is a minor semantic point and does not detract from the method's novelty.

**Questions:**

Please refer to the weakness part.

---

> ### Author Response · Authors · 2025-11-20
>
> ​​Thank you for your time and your positive review of our work. Below, we address your questions and the weaknesses mentioned.
>
> > **Dependency on Teacher Model Quality**
>
> We observe that our model is very robust to both generated and real input images, as we do not cherry-pick our results. Initially, we manually inspected the data generation outputs and found that overall quality was good; therefore, we did not perform any data filtering when training our 3DGS decoder from the video model's output, demonstrating the approach’s ability to handle diverse inputs. The dependency on the teacher model is actually beneficial, as the 3D consistency and visual quality of video models are continually improving. These orthogonal improvements to the base video model will be directly applicable to our approach, resulting in visual and quality improvements in the 3D output.
> Designing data filtering mechanisms for our distillation framework, e.g., rejecting video samples that are inconsistent or contain artifacts, potentially using vision-language models (VLMs), is an interesting direction for future work and could further improve visual quality. All quantitative comparisons demonstrate that our model outperforms previous work and effectively demonstrates the approach’s effectiveness and robustness. We will add further discussion of data filtering and the teacher model's dependency in the paper.
>
> > **Reliance on External Depth Estimation**
>
> While the depth loss provides a slight improvement in visual quality and 3D consistency, its contribution is modest and not essential to the performance of our main loss, as shown in the Table below:
>
> *Table: Ablation for the depth loss*
>
> |Method|PSNR|SSIM|LPIPS|
> |:--------|:--------:|:--------:|:--------:|
> |Ours|**24.77**|**0.837**|**0.224**|
> |w/o depth loss|24.31|0.811|0.247|
>
> The visual ablations in Fig. 11 (Appendix, page 26) support these observations. Even without the depth loss, the 3DGS renderings and depth maps maintain high quality. Depth supervision offers a minor enhancement, but it can be omitted without causing a noticeable drop in results. As video models become increasingly 3D-consistent, we expect the effect of depth loss to become even less significant.
>
> > **Minor Terminology Nitpick**
>
> This is a great point, and we agree that our setup is not a typical self-distillation framework, as the modalities and architectures are changing. We would like to get your opinion on which of the following changes is more suitable. One solution is to rename self-distillation to cross-modal distillation throughout the paper. The alternative solution is to add a clarification statement in the paper upfront, stating that we do not propose a classic self-distillation framework but rather use different architectures for the teacher and student within a cross-modal distillation framework. This alternative solution would enable us to retain the current paper title and provide a clearer explanation of the distillation setup in the paper, similar to this:
>
> “While we refer to our training procedure as “self-distillation”, our framework more precisely constitutes a cross-modal distillation process, in which supervisory signals are transferred from the VAE’s 2D RGB decoder to our newly introduced 3DGS decoder.”
>
> We are happy to discuss this terminology further with you to arrive at a suitable solution before updating the paper.

---

### Meta-Review · Area_Chair_ftqY · 2026-01-04

**Summary:**

The paper proposes to generate 3D/4D Gaussian Splatting scenes from monocular input. It introduces and trains a 3D-GS decoder using the latent features and RGB supervision from a camera-conditioned video diffusion model. Initially, it receives 4 reviews in total (two 8 ratings and two 6 ratings). The reviewers acknowledged 1) its novel self-distillation approach; 2) SOTA performance and representation; 3) it provides extensive experiments etc. The reviewers raised several concerns regarding 1) its dependency on teacher model quality; 2) reliance on external depth estimation; 3) limited technical contributions; 4) temporal/multi-view inconsistencies; 5) efficiency on the training process etc. The authors provide detailed rebuttal and addressed most of the critical concerns. After carefully weighing the paper's strengths against its limitations, the AC is happy to recommend the acceptance of the paper.

**Reviewer Concerns:**

The concerns are addressed by the rebuttal.

**Reviewer Scores:**

Based on the initial reviews, authors rebuttal and initial discussion between the authors and the reviewers, the AC think the reviewers would keep their original ratings.

---

### Decision · Program_Chairs · 2026-01-26

Accept (Poster)